# Promoting Mental Health and Well-Being Among Adolescent Young Carers in Europe: A Cross-National Randomized Controlled Trial Study

**DOI:** 10.3390/healthcare12212124

**Published:** 2024-10-24

**Authors:** Valentina Hlebec, Irena Bolko, Giulia Casu, Lennart Magnusson, Licia Boccaletti, Renske Hoefman, Alice De Boer, Feylyn Lewis, Agnes Leu, Francesco Barbabella, Rosita Brolin, Sara Santini, Marco Socci, Barbara D’Amen, Daniel Phelps, Tamara Bouwman, Nynke de Jong, Elena Alder, Vicky Morgan, Tatjana Rakar, Saul Becker, Elizabeth Hanson

**Affiliations:** 1Faculty of Social Sciences, University of Ljubljana, Kardeljeva pl. 5, 1000 Ljubljana, Slovenia; irena.bolko@fdv.uni-lj.si (I.B.); tatjana.rakar@fdv.uni-lj.si (T.R.); 2Department of Psychology «Renzo Canestrari», University of Bologna, Viale Berti Pichat 5, 40127 Bologna, Italy; giulia.casu3@unibo.it; 3Department of Health and Caring Sciences, Linnaeus University, SE-39182 Kalmar, Sweden; lennart.magnusson@lnu.se (L.M.); francesco.barbabella@lnu.se (F.B.); rosita.brolin@lnu.se (R.B.); elizabeth.hanson@lnu.se (E.H.); 4The Swedish Family Care Competence Centre (NKA), Strömgatan 13, SE-39232 Kalmar, Sweden; 5Anziani e non solo soc. Coop. Sooc, Via Lenin 55, 41012 Carpi, Italy; progetti@anzianienonsolo.it; 6The Netherlands Institute for Social Research (SCP), Postbus 16164, 2500 BD Den Haag, The Netherlands; r.hoefman@scp.nl (R.H.); a.de.boer@scp.nl (A.D.B.); 7Vanderbilt University School of Nursing, 461 21st Ave South|#179 SON, Nashville, TN 37240, USA; feylyn.m.lewis@vanderbilt.edu; 8Institute for Biomedical Ethics, Medical Faculty, University of Basel, 4056 Basel, Switzerland; agnes.leu@unibas.ch; 9Centre for Socio-Economic Research on Aging, IRCCS INRCA—National Institute of Health and Science on Aging, Via Santa Margherita 5, 60124 Ancona, Italy; s.santini2@inrca.it (S.S.); m.socci@inrca.it (M.S.); 10Italian National Institute of Statistics—ISTAT, 00184 Rome, Italy; barbara.damen@istat.it; 11Careum School of Health, Kalaidos University of Applied Sciences, Gloriastrasse 18a, 8006 Zurich, Switzerland; daniel@youngcarers.info (D.P.); elena.alder@careum-hochschule.ch (E.A.); 12Vilans—The National Centre of Expertise for Long-Term Care in The Netherlands, Churchilllaan 11, 3527 GV Utrecht, The Netherlands; t.bouwman@vilans.nl (T.B.); n.p.dejong@utwente.nl (N.d.J.); 13Carers Trust, 2 6 Boundary Row, London SE1 8HP, UK; vmorgan@carers.org; 14Institute for Children’s Futures, Manchester Metropolitan University, Manchester M15 6BH, UK; s.becker@mmu.ac.uk

**Keywords:** adolescent young carers, psychosocial support, primary prevention, mental health, intervention study, cross-national study, randomized controlled trial

## Abstract

Background/Objectives: This cross-national study focuses on adolescents who provide care and support to family members or significant others. Current evidence regarding their mental health and solutions to strengthen it is limited and mostly available in a few countries. The aim of this study is to evaluate the results of a primary prevention intervention for improving the mental health and well-being of adolescent young carers (AYCs) aged 15–17 years in six European countries. The intervention was based on a psychoeducational program and tools adapted from the Discoverer, Noticer, Advisor, and Values (DNA-V) model. Methods: We designed a randomized controlled trial with 217 AYCs participating in the study, either in the intervention or control group. Quantitative and qualitative data were collected via questionnaires at baseline, post-intervention, and a 3-month follow up. Results: The results were mixed, as positive improvements in primary (i.e., psychological well-being and skills) and secondary (school/training/work functioning) outcomes were shown by the experimental group but, in most cases, they were not statistically significant. The qualitative data supported positive claims about the intervention and its appropriateness for AYCs. Conclusions: The study implementation during the peak of the COVID-19 pandemic forced the consortium to adapt the design and may have influenced the results. More long-term studies are needed to assess similar mental health programs with this hard-to-reach target group.

## 1. Introduction

Caring responsibilities in children and young people may have an adverse impact on psychological adjustment and challenge the reconciliation of caring responsibilities with education, career, and personal life [1]. Although these figures may be underestimations, the available data suggest that around 4–8% of children and young people aged between 10 and 24 years in the EU take on caring duties for family members [2,3]. Assuming caring responsibilities in a developmental phase has especially been connected to a negative impact on psychological well-being [4,5,6,7]. Education can be challenging for adolescent young carers (AYCs), which negatively impacts their employability and subsequently their socioeconomic status [8,9,10]. Regardless of the negative consequences of caring at a young age, evidence of effective psychological interventions is scarce [11,12] and limited to specific national settings. In this study, the evaluation results of a primary prevention intervention for AYCs aged 15–17 tested in six European countries (Italy, Slovenia, Sweden, Switzerland, the Netherlands, and the United Kingdom) are presented. The intervention was developed in co-design with professionals and young carers within the research project “Psychosocial support for promoting mental health and well-being among adolescent young carers in Europe” (ME-WE), funded by the European Union (2018–2021) [13]. The ME-WE primary prevention intervention program is aimed at promoting the mental health and well-being of AYCs, and it is an adaptation of the theoretical framework of the DNA-V model (Discoverer, Noticer, Advisor, and Values) to the specific experiences of adolescents with caring responsibilities [14]. This framework was chosen following the research conducted in an earlier study of the ME-WE project, where a key finding was that one of the most promising models for use with AYCs was The Resourceful Adolescent Program (RAP-A). There is indeed an identified lack of evidence-based support interventions targeting AYCs that focus on promoting mental health and well-being. The RAP-A model addresses the same psychological processes as the DNA-V. However, the DNA-V model was preferred because it is shorter (7 sessions vs. 11 sessions for RAP-A) and is based on the third generation of cognitive behavioral therapy (while RAP-A is based on the previous generation, and it is therefore an older model). Moreover, RAP-A was specifically conceived to prevent depression, while DNA-V was not developed to tackle a specific disease but rather to enrich young people’s lives and fundamentally transform the way they handle difficult thoughts, feelings, and situations.

The objective of the ME-WE primary prevention intervention was to help AYCs recognize, accept, and share their emotions; to improve psychological flexibility, mindfulness, resilience, and mental and physical health; to increase caring-related quality of life; and develop new, effective ways of balancing a caring role and social relationships among AYCs [13,15]. The primary outcomes targeted by the ME-WE intervention included psychological flexibility, mindfulness, resilience, mental and physical health, impact of caring, and social support. Secondary outcomes not directly addressed by the ME-WE intervention included self-reported functioning at school/training/work.

This article presents a cross-national analysis of the evaluation of the ME-WE primary prevention intervention using a mixed-methods design and aiming at establishing the impacts of the ME-WE primary prevention intervention on the mental health and well-being of AYCs.

## 2. Materials and Methods

### 2.1. Study Design

This study is a randomized controlled trial (RCT) with a two (arms) by three (times) repeated-measures factorial design. Fieldwork was conducted in six European countries (Italy, Slovenia, Sweden, Switzerland, the Netherlands, and the United Kingdom). The details of the full trial methodology are described in the ME-WE study protocol article [15]. The intervention was mostly implemented during the COVID-19 pandemic (from March 2020 to April 2021), which imposed considerable challenges on the clinical trial study in all six countries. To comply with the restrictions and precautionary measures introduced at national levels, the study was virtualized in the spring of 2020. However, no changes in the intervention contents «per se» were introduced, keeping the online intervention equivalent to the pre-COVID-19 version.

Cluster randomization was adopted by Italy, the Netherlands, Slovenia, and the United Kingdom. Clusters consisted of AYCs attending the same school (in Slovenia) or AYCs or organizations (schools/care support centers) living or located in the same geographical area (i.e., neighborhood or post-code area) (in Italy, the Netherlands, and the United Kingdom). Due to COVID-related challenges in recruitment, Sweden and Switzerland planned the randomization of individual participants who were recruited via national social media recruitment campaigns and through schools and stakeholders. Clusters or individual participants were randomly assigned to either the ME-WE intervention or the waitlist control group.

Two delivery methods were applied for the two groups of countries in the RCT. The main difference between the delivery modes was that one mode used only face-to-face methods, while the other mode also incorporated online methods for facilitating sessions where participants met. While both approaches worked on the same processes, had the same goals, and were based on similar exercises and activities, they were conceived to respond to different national/regional characteristics in terms of the propensity of the target groups to use information and communication technologies, and the level of awareness of young carers’ issues. The first mode, named the fully face-to-face approach (FTF), was implemented in three countries (Italy, Slovenia, and the United Kingdom). The second delivery mode combined face-to-face methods with an online tool for facilitating training sessions. In this so-called blended approach, the online training sessions were delivered using an online communication system, ZOOM, and a dedicated ME-WE mobile app (APP) specifically co-designed with AYCs in an earlier study within the ME-WE project. In the blended approach, four out of eight sessions of the program were performed online, with the theoretical inputs provided through short videos and exercises/activities adapted to be performed by participants online and remotely. Three countries (Sweden, Switzerland, and the Netherlands) used the app.

The evaluation of the intervention employed a robust mixed-methods triangulation design [16], where the quantitative survey questionnaire was complemented with open-ended questions aimed at gathering AYCs’ impressions of intervention activities and the intervention itself, especially after employing COVID-19 social distancing, as well as limitations in implementing the intervention. The quantitative survey questionnaire was filled in by participants in the training sessions (ME-WE intervention and waitlist control group). Instruments were used that measured outcomes at the individual level. Data were collected at three moments: First, at baseline (T0). The second measurement (T1) was conducted after the last training session for the ME-WE intervention group. Data for the control group were collected 7 weeks after T0 for the waitlist control group. The third, and last, measurement (T2) was conducted at 3 months after T1 for both the ME-WE intervention and the control group. After filling in the third survey, participants in the control group were invited to participate in the training program that was previously offered to the intervention group. The Consolidated Standard of Reporting Trials (CONSORT) 2010 statement extended to cluster trials is followed in this study. The research flow diagram is presented in Figure 1.

The trial was registered at clinicaltrials.gov (trial registration number: NCT04114864). To ensure compliance with the agreed study protocol and to maintain the highest research ethics standards, country clinical trial managers (CTMs) were appointed within each of the six partner countries and regular meetings were held, chaired by the overall project CTM (LM). In keeping with their respective national human ethics legislation and ethics approval procedures, Italy, Slovenia, the Netherlands, the United Kingdom, and Sweden sought and obtained formal ethical approval from their relevant Ethical Review boards. Namely, from the University of Bologna, University of Ljubljana, Vrije Universiteit Amsterdam, University of Sussex, and the Swedish Ethical Review Authority. In Switzerland, in accordance with the national human research act, the Ethics Committee of the Zürich Canton deemed formal ethical approval unnecessary. Nevertheless, detailed opinions were sought and secured from the Committee by the Swiss research team. During the COVID-19 outbreak, amendments had to be made to the study design. These amendments were reviewed for ethical aspects in all countries. This resulted in formal ethical approval (Italy, Slovenia, the Netherlands, the UK, Sweden) and detailed opinions (Switzerland), in accordance with national human ethics legislation (see also the Institutional Review Board Statement in this paper).

All eligible potential AYC participants received an information letter and an informed consent form. All enrolled AYCs signed or provided their verbal consent prior to participating in the study. Due to COVID-19 restrictions, informed consent was provided in written form digitally (IT, SI, NL, CH), by postal mail (IT), or orally in a recorded online meeting (UK, SE). Parental consent was secured in Italy, the Netherlands, Slovenia, and the UK in accordance with the respective national legal ages of consent (15–17 years, IT; 15 years, NL and SI) or mandated national good practice (UK; 16–17-year-olds, NL). In Sweden and Switzerland, parental consent is not a legal requirement for young people aged 15–17 years. Thus, in Switzerland, enrolled AYCs were asked prior to the first session if they wished to receive (via email) an information letter for their parents/guardians. In Sweden, potential participants were sent a separate email with an information letter for their parents, which they were asked to forward to their parents/guardians. In keeping with national legal requirements, parental consent was secured in Switzerland and Sweden for compassionate cases where participants were 14 years old.

### 2.2. Participants

#### 2.2.1. Inclusion and Exclusion Criteria

To be eligible to participate in the study, participants had to meet the following inclusion criteria: (1) being in the age group between 15 and 17 years; (2) caring for one or more persons close to them, namely family member(s) (e.g., parents, siblings, or grandparents) or significant other (e.g., friend, partner, schoolmate, or neighbor) with a health-related issue such as chronic physical and/or mental health illness or substance abuse, neurodevelopmental or neurological disorders, a disability and/or other health problems related to old age [17,18]. Furthermore, potential participants were assessed for the following exclusion criteria: (1) concurrently participating in other mindfulness-based interventions or programs or taking psychotherapy sessions; (2) having started to take a new psychotropic medication within the past month, or, alternatively, planning on starting or changing psychotropic medication during the participation in the present study/intervention; (3) limited knowledge of the local language where the intervention is taking place, with the exception of Sweden, where it was planned to also include in the study AYCs with limited knowledge of Swedish.

Based on ethical and compassionate grounds, a small number of AYCs who did not meet the inclusion criteria were also accepted to take part in the intervention. They expressed a need for support and interest in participation, and no other support alternatives comparable to the ME-WE groups were available in their region at the time of the intervention study. Especially during the COVID-19 pandemic, the support of these compassionate cases became even more relevant, because all regular support for young people in all countries was canceled or reduced due to the pandemic restrictions. For these compassionate cases, participation in the ME-WE groups was allowed only after a case-by-case assessment carried out by the National Clinical Trial Manager (CTM) and Ethics, Gender and Data Manager (EGDM) [13].

#### 2.2.2. Sample Size

The procedure for computing the minimal required sample size is presented in the study protocol article [15]. Target sample sizes for each country were between 58 and 142, depending on the type of recruitment design (cluster- vs. individual-based).

#### 2.2.3. Recruitment

Actual sample sizes were significantly reduced compared to planned sizes, because of challenges encountered in the recruitment process and the restrictions imposed by COVID-19 pandemic protection measures [19]. Sample sizes were as follows: 45 AYCs (27 AYCs in the intervention and 18 AYCs in the waitlist group) in Italy, 24 AYCs (16 AYCs in the intervention and 8 AYCs in the waitlist group) in both the Netherlands and Sweden, 82 AYCs (33 AYCs in the intervention and 49 AYCs in the waitlist group) in Slovenia, 4 AYCs (1 AYC in the intervention and 3 AYCs in the waitlist group) in Switzerland, and 38 AYCs (15 AYCs in the intervention and 23 AYCs in the waitlist group) in the United Kingdom. The total sample size was 217, but due to small sample sizes in Switzerland, Swiss participants were excluded from the quantitative data analysis, resulting in 213 analyzed assessments (as reported in Figure 1).

Intervention group membership was reduced by excluding 43 participants who attended less than 50% of the sessions prior to the statistical analyses (and considered dropouts, which were not included in the above reported sample sizes; see Figure 1, where dropouts are reported under ‘other reasons’).

Differences across countries were sizeable; however, a common tread was that recruitment for control groups was often delayed and resulted in an uneven distribution of participants in the intervention and control groups.

In order to boost recruitment, a variety of recruitment methods were employed, such as including presentations of the research project in schools and youth centers, promotional materials such as leaflets and posters, social networks, media and press releases, and healthcare professionals’, social workers’, and teachers’ referrals [19].

The eligibility of participants was assessed by research team members with a brief face-to-face or telephone screening, where AYCs were encouraged to ask questions about the project. Remote recruitment and screening were introduced in all countries after the COVID-19 pandemic outbreak.

### 2.3. Intervention

The ME-WE primary prevention intervention was designed by adapting the existing DNA-V protocol [14] to meet the specific needs and experiences of AYCs aged 15–17. This adaptation was developed through blended learning networks involving young carers and professionals across six countries [13]. Participants in the ME-WE intervention group attended seven group sessions, each lasting approximately two hours and typically held weekly, with a follow-up session three months after the intervention concluded. Groups were comprised of 2 to 9 participants in most of the cases. Group dynamics represented an important part of the intervention. In small- sized groups, facilitators supported group dynamics by adopting a participant role when needed. All sessions followed a similar structure (objectives, icebreaker, central activity/ies, and final activity) [15].

The DNA-V model is a psychological intervention, addressed to adolescents and young people, used in educational and clinical settings. This model has its roots in contextual and functional science and is based on Acceptance and Commitment Therapy [20], a third-generation cognitive behavioral therapy. The model defines three functional classes of behavior—Discoverer, Noticer, and Advisor (DNA)—with Values (Vs) at the core to guide behaviors. The Discoverer represents exploratory behaviors. The Noticer is a powerful process that allows adolescents to connect with their feelings, body, and the physical signals coming from the world around them. The Advisor is concerned with how adolescents use their inner voice or self-talk to make sense of the past, form beliefs, evaluate themselves, and predict the future. Values can be thought of as a compass that guides people through the storms and confusing times of life and toward the things they care about.

Through the seven sessions on which the event is based, participants are accompanied to explore the DNA behaviors and to understand the role of Values through a series of group dynamics and exercises connected with their role as informal carers. Both delivery approaches (FTF and APP) addressed the same psychological processes and used similar exercises and activities. In the blended approach, sessions 1, 3, 7, and the 3-month follow-up meeting were conducted face-to-face, while sessions 2, 4, 5, and 6 were conducted online using the ME-WE app and the ZOOM platform [15]. In response to the COVID-19 pandemic, both delivery methods were adopted to accommodate full online delivery. Regardless of the adaptation, the FTF and APP deliveries were preserved. All changes made to the intervention program were conservative, only allowing the transfer to a virtual environment. Meta-analytic evidence speaks in favor of the equivalence of face-to-face and web-based psychological interventions [21,22].

For the control group, AYCs participated in icebreaker and team building activities during three meetings aligned with the three assessment points to collect outcome measures. Prior to March 2020, these meetings were face-to-face, but they transitioned to online formats using video-conferencing software due to the pandemic. Control group participants were offered the ME-WE intervention after the experimental group had completed all the assessments, including the 3-month follow-up.

In both delivery approaches, one or two facilitators conducted each session. All facilitators received theoretical and practical training in the DNA-V model and intervention exercises, as well as a DNA-V protocol manual and/or training course [https://www.praxiscet.com/dnav-for-young-people-evergreen-signup/ (accessed on 30 June 2024)] and the ME-WE intervention manual. Facilitators were recruited amongst psychologists, youth workers, teachers, school nurses, and social workers. Prior experience with group work and young people was deemed advantageous. Debriefing sessions were offered to facilitators throughout the study duration by a project member trained in the DNA-V model.

### 2.4. Study Outcomes

All the validated self-reported measures selected for this study were specifically designed for and/or tested with adolescent samples. For tools not available in the languages of the participating countries, two independent researchers per country translated them following the standard committee procedure [23], after obtaining permission from the original authors.

Both web-based and paper-and-pencil versions of the questionnaire were prepared. After the outbreak of the COVID-19 pandemic, only the web-based version of the study questionnaires was used, and additional COVID-19-related measures were included to evaluate participants’ experiences with the adapted intervention and the perceived impact of the pandemic on their health and caregiving role.

A sociodemographic form was administered at T0, asking for gender, age, country of birth, nationality, migration history, area of living, family composition, and living conditions. Caregiving-related information was also collected at T0, including the number and age of people cared for, the relationship with the individual receiving care, the type of health-related condition of the individual receiving care, and the duration of caregiving.

#### 2.4.1. Primary Outcomes

Psychological flexibility was assessed using the Avoidance and Fusion Questionnaire for Youth (AFQ-Y) [24]. The AFQ-Y measures psychological inflexibility, defined as the tendency to avoid unpleasant thoughts and feelings while becoming entangled with them, hindering adaptive functioning and goal-directed behavior in youth. It includes 8 items (e.g., “My thoughts and feelings mess up my life”) rated on a 5-point scale from ‘not at all true’ to ‘very true’. Higher total scores indicate higher psychological inflexibility. Cronbach’s α coefficients at baseline were 0.74 (IT), 0.79 (NL), 0.76 (SE), 0.78 (SI), and 0.86 (UK).

Mindfulness skills were assessed using the Child and Adolescent Mindfulness Measure (CAMM) [25]. The CAMM assesses mindfulness in youth, defined as the ability to be aware of the present moment with a non-judgmental attitude, promoting emotional regulation and reducing reactivity to thoughts and feelings. It consists of 10 items (e.g., “I keep myself busy so I don’t notice my thoughts or feelings.”) rated on a 5-point scale from ‘never true’ to ‘always true’. An overall mindfulness score is obtained by reverse scoring the items and summing the total, so that higher scores indicate greater mindfulness. Cronbach’s α coefficients at baseline were 0.74 (IT), 0.85 (NL), 0.67 (SE), 0.75 (SI), and 0.87 (UK).

Resilience was assessed using the Brief Resilience Scale (BRS) [26]. The BRS includes 6 items (e.g., “I usually come through difficult times with little trouble”) describing an individual’s ability to recover from stress and adversity, focusing on their capacity to bounce back and maintain psychological stability in challenging situations. Items are rated on a 5-point scale from ‘strongly disagree’ to ‘strongly agree’. After reverse scoring negatively worded items, a total score is obtained, with higher scores indicating higher levels of resilience. Cronbach’s α coefficients at baseline were 0.69 (IT), 0.81 (NL), 0.74 (SE), 0.64 (SI), and 0.78 (UK).

Mental well-being was assessed using the Warwick Edinburgh Mental Well-Being Scale (WEMWBS) [27]. The WEMWBS measures mental well-being by assessing positive aspects of mental health, including optimism, positive relationships, and overall life satisfaction, capturing a holistic view of an individual’s psychological well-being. Its 14 items (e.g., “I’ve been thinking clearly”) are rated on a 5-point scale from ‘none of the time’ to ‘all of the time’, referring to the individual’s experience over the past two weeks. Higher total scores indicate a higher level of mental well-being. Cronbach’s α coefficients at baseline were 0.88 (IT), 0.83 (NL), 0.87 (SE), 0.91 (SI), and 0.91 (UK).

Quality of life was assessed using the KIDSCREEN-10 [28], which includes 10 items (e.g., “Have you felt fit and well?”) capturing various dimensions of well-being—such as physical, emotional, mental, social, and behavioral aspects—over the past week, to provide a comprehensive overview of their overall quality of life. Items are rated on 5-point scale from ‘not at all/never’ to ‘extremely/always’, and higher total scores indicate a better quality of life. Cronbach’s α coefficients at baseline were 0.74 (IT), 0.79 (NL), 0.73 (SE), 0.79 (SI), and 0.77 (UK).

Subjective health complaints were assessed using the HBSC Symptom Checklist (HBSC) [29]. The HBSC consists of 8 items addressing the frequency of physical and psychological health complaints, such as headaches, stomach aches, irritability, and feeling low, during the previous six months. Items are rated on a 5-point scale from ‘almost every day’, to ‘rarely or never’. Higher total scores indicate better psychosomatic health, meaning fewer reported complaints. Cronbach’s α coefficients at baseline were 0.87 (IT), 0.68 (NL), 0.76 (SE), 0.79 (SI), and 0.85 (UK).

Caring-related quality of life was evaluated using three closed-ended Yes/No questions developed specifically for this study. These questions asked whether participants had experienced thoughts of self-harm due to their caregiving activities, thoughts of harming others, or if they had been bullied, teased, or made fun of because of their caring responsibilities. Additionally, an ad hoc multiple-choice question was included to assess whether participants had any health-related issues as a result of their caring role (i.e., mental health problems, physical disabilities, learning difficulties, or other health-related conditions).

The cognitive and emotional impact of caregiving was measured using the Positive and Negative Outcomes of Caring (PANOC) [30,31], which includes two 10-item subscales addressing positive (e.g., “Because of caring I feel that I am learning useful things”) and negative outcomes (e.g., “Because of caring I can’t stop thinking about what I have to do”) of caring. Items are rated on a 3-point scale ranging from ‘never to ‘a lot of the time’. For each subscale, a total score can be computed, with higher scores reflecting greater positive or negative outcomes of caregiving. At baseline, Cronbach’s α coefficients were 0.83 (IT), 0.90 (NL), 0.87 (SE), 0.90 (SI), and 0.85 (UK) for positive outcomes, and 0.91 (IT), 0.85 (NL), 0.85 (SE), 0.75 (SI), and 0.85 (UK) for negative outcomes.

#### 2.4.2. Secondary Outcomes

Self-reported experiences in school, training, or work, including attendance (e.g., I miss school/training/work) and performance (e.g., “It is difficult for me to perform well at school/training/work”), were measured using 5 ad hoc items that inquired about perceived caring-related difficulties in these areas. Items were rated on a 5-point scale ranging from ‘never’ to ‘always’. A total score was computed by summing the 5 items, with higher scores indicating greater caring-related difficulties in school/training/work. Cronbach’s α coefficients at baseline were 0.65 (IT), 0.64 (NL), 0.73 (SE), 0.65 (SI), and 0.75 (UK). At T2, 3 ad hoc items were used to assess the likelihood of continuing or completing education/training compared to before participating in the study, and whether the intervention influenced this decision. Items were rated on a 5-point scale ranging from ‘yes, absolutely’ to ‘absolutely not’.

#### 2.4.3. Additional Measures

The extent of caring activities was evaluated using the Multidimensional Assessment of Caring Activities (MACA) [30,31], which consists of 18 items rated on a 3-point scale ranging from ‘never’ to ‘a lot of the time’. The MACA assesses various aspects of caregiving, including domestic tasks, household management, personal care, emotional support, sibling care, and financial/practical assistance. Subscale and total scores can be computed, with higher scores reflecting a greater amount of caring activity undertaken by the AYC. Cronbach’s α coefficients for the total scale at baseline were 0.83 (IT), 0.59 (NL), 0.76 (SE), 0.79 (SI), and 0.78 (UK).

Likes and dislikes about caring were evaluated through four open-ended ad hoc questions that asked AYCs which caring tasks they enjoyed the most, disliked the most, found most gratifying, and found most upsetting.

Three open-ended questions were used to ask AYCs about the impact of the COVID-19 pandemic on their lives, mental health, and physical health, as well as whether they or their families were receiving the necessary support and services during the COVID-19 crisis. AYCs in the intervention group were also asked an additional open-ended question about their experiences of participating in the ME-WE sessions during the pandemic and their engagement with the exercises and home activities proposed in the ME-WE intervention. These questions were administered at both T1 and T2.

Additionally, all participants evaluated their satisfaction with the overall support they received on a 5-point scale, ranging from very satisfied to very dissatisfied.

#### 2.4.4. PISA—Post-Intervention Self-Assessment (PISA)

Immediately after the intervention, AYCs in the ME-WE intervention group were asked to complete an adapted version of the Post-Intervention Self-Assessment by Joseph et al. [31]. This assessment included 7 dichotomous Yes/No items related to their experiences with the ME-WE intervention (e.g., “I enjoyed most of the activities”), 10 items about changes related to participation (e.g., “I feel able to choose the level of care I provide”) rated on 3-point scale (ranging from ‘more often than before the intervention’ to ‘less often than before the intervention’), and 5 open-ended questions. The open-ended questions asked AYCs to describe their perceptions of the support received from the intervention, any changes they experienced in their caring role as a result of participating, aspects of the intervention they did not like, and any additional feedback they wished to share with the research team.

#### 2.4.5. Evaluation of Online Delivery of the ME-WE Intervention

Participants in the Netherlands, Sweden, and Switzerland were also asked two open-ended questions regarding their experience with the ME-WE app and suggestions for its improvement.

### 2.5. Data Analysis

#### 2.5.1. Statistical Analysis

Data were analyzed using intention-to-treat principles [32]; thus, all participants that completed the first (baseline) assessment and attended at least half of the sessions were included in the analysis. Missing values on the scales were imputed by the maximum expectation method [33]. Sociodemographic variables, variables referring to caring situation (e.g., persons who AYCs are providing care and support for), and questions with a Yes/No response option were not imputed.

At baseline, equivalence was checked between study groups (ME-WE FTF intervention, ME-WE APP intervention, and waitlist control) using ANOVA (with Tukey’s HSD post hoc test for multiple comparisons) and *χ*^2^ tests or Fisher’s exact test in the case of low (i.e., <5) frequencies (with post hoc *z*-test). Differences in outcomes between the groups from baseline (T0) through post-intervention (T1) and 3-month follow-up (T2) were tested using mixed ANOVA, including fixed effects of time and time–group interactions. Due to previously described recruitment difficulties that resulted in largely unrealized and uneven sample sizes, we excluded exploring possible country effects in the data analysis.

Statistical analyses were based on individual participant-level data. It should be noted that it was not feasible to include cluster-based adjustments [34,35]. Due to the COVID-19 pandemic, Sweden and Switzerland endeavored to turn to an individual-based RCT. According to Dreyhaupt [35] (p. 7), individual-level analysis is an alternative approach to cluster-level analysis that is more efficient for strongly varying cluster sizes, as was the case in the present study. Due to the small sample size in Switzerland, the Swiss quantitative data were not included in the international comparative analysis. The interpretation of results was based on both statistical significance (*p* < 0.05, two-tailed with Bonferroni correction for multiple testing where applicable) and measures of effect size [36]. The analysis was conducted using IBM SPSS Statistics 25.

#### 2.5.2. Qualitative Analysis

To supplement the quantitative data analysis and gain a more in-depth understanding of AYCs’ subjective experiences of the intervention and impact of the COVID-19 pandemic on their situation, the research teams performed a content analysis of their national qualitative data, which also included the Swiss qualitative data. This was collected from AYCs’ answers to the open-ended questions of the evaluation questionnaire, using a code tree developed by the research team. Additionally, some of the teams added codes specific to their respective country if the data analysis revealed country-specific topics. The qualitative findings presented in this article only concern the findings from the intervention groups.

## 3. Results

### 3.1. Sample Characteristics

Table 1 presents sociodemographic characteristics of the ME-WE FTF and ME-WE APP groups as well as those of the waitlist group. The mean age of all participants was 16.36 (SD = 0.87) years. Most participants were female, 79.2%, while the whole sample included 17.5% males, and 3.3% of participants identified themselves as transgender, another gender, or they did not disclose their gender. Most of the participants lived in a town or a small city (45.0%), followed by a country village (21.8%). Participants living in a big city or in the suburbs or outskirts of a big city were less prevalent (15.6% and 12.8%, respectively). Only a few participants reported living on a farm or in the countryside (4.7%). The whole sample included 9.0% who were born in another country (rather than the country they currently lived in). For 15.6% of participants, their mother was born in another country, while 16.1% of fathers were born in another country.

In total, 94.8% of participants stated that they lived with their mother, 72.5% of participants lived with their father, 2.4% of participants lived with their stepmother (or parent’s girlfriend), and 5.7% with their stepfather (or parent’s boyfriend). Participants also reported that their household included brother(s) (46.4%), sister(s) (51.2%), grandmother (14.2%), and grandfather (7.6%). Additionally, 1.4% of participants lived in a foster home and 0.9% lived in a children’s home, while 4.7% of participants named someone or somewhere else.

A breakdown of participants’ characteristics for the three groups is presented in Table 1. There were no statistically significant differences in sociodemographic variables, except for living with a mother or with someone or somewhere else. Post hoc *z*-tests with Bonferroni adjusted alpha levels of 0.017 indicated a statistically significant different share of participants not living with their mother in the APP group in comparison to the FTF and waitlist group (*z* = −2.9, *p* = 0.004). In the APP group, there was also a larger share of participants living with someone or somewhere else, but this difference was not significant when applying the Bonferroni correction (*z* = 2.2, *p* = 0.028).

Participants were also asked about the family members and close friends with the health-related condition that they look after, help, or support. The results are presented in Table 2 (family members) and Table 3 (close friends).

In total, participants named 242 family members who needed help or support. Of these, 70 family members were named in the ME-WE FTF group, 47 in the ME-WE APP group, and 125 in the waitlist group. In 27.2% of cases, participants named their mother, in 10.7% of cases their father, in 15.6% and 12.8% of cases their sister or brother, respectively, in 14.0% of cases their grandmother, in 6.6% their grandfather, in 2.9% their cousin, in 2.9% their uncle, in 2.5% their aunt, and in 4.5% of cases participants named other family members in need of help or support. The mean age of all family members in need of care was 42.65 years (*SD* = 23.68, range 3–90) and in 68.2% of cases the participants lived with them. For a breakdown of the results by group, see Table 2. When applying post hoc *z*-test with Bonferroni adjusted alpha levels of 0.017, we found two statistically significant differences among the groups, namely in the share of participants who listed their brother in the APP group (*z* = 2.4, *p* = 0.016) and in the share of participants who named their cousin in the waitlist group (*z* = 2.6, *p* = 0.009). In addition, the mean age of family members in the APP group was lower compared to the waitlist group (mean difference 95% CI [−18.90, 0.05], but *p* = 0.05 exceeded the Bonferroni corrected significance threshold of 0.017).

In 30.8% of the cases, family members had physical illness (for example, cancer, diabetes, asthma, lung disease, heart disease, or other kinds of physical illness), followed by cognitive impairments (for example, cognitive impairments caused by dementia or Alzheimer’s, autism, learning disorders, traumatic brain injury, Down’s Syndrome, attention deficit hyperactivity disorder (ADHD), or other kinds of cognitive impairments) in 20.9% of the cases, ill mental health (for example, depression, fatigue syndrome, bipolar disorder, anxiety, phobias, obsessive compulsive disorder, borderline, post-traumatic stress syndrome, psychosis, schizophrenia, self-harming, suicidal thoughts, eating disorder, or other kinds of ill mental health) in 19.7% of the cases, and physical disability (for example, physical disabilities caused by frailty, accident, injury, illness, or other causes) in 19.1% of the cases. Addiction (for example drugs, alcohol, gambling, or other kinds of addiction) was reported in 2.8% of the cases, while in 6.8% of the cases, other health-related conditions were named. Multiple answers were possible with this question. For a results breakdown by group, see Table 2. In the APP group, there was a statistically significant lower share of participants who reported that their family members had physical illness (*z* = −2.9, *p* = 0.004), but a higher share reported cognitive impairments (*z* = 2.4, *p* = 0.016).

Participants named 145 close friends in need of help or support. Of those, 32 close friends were named in the ME-WE FTF group, 30 in the ME-WE APP group, and 83 in the control group. In 80.9% of the cases, they were friends, in 8.8% girlfriends or boyfriends, in 8.1% colleagues, in 3.7% neighbors, in 2.2% ex-girlfriends or ex-boyfriends, and in 3.7% of the cases they were other close people. The mean age of these persons was 18.86 (*SD* = 10.81, range 9–74), and in 3.5% of cases the participants lived with them. For a results breakdown by group, see Table 3. In the FTF group, there was a significantly higher share of participants who lived with the person in need of help or support (*z* = 3.1, *p* = 0.002), while in the APP group there was a significantly higher share of participants who named their girlfriend or boyfriend (*z* = 2.6, *p* = 0.009).

In 57.5% of the cases, close friends were reported to have ill mental health, followed by physical illness or cognitive impairments, both in 11.7% of the cases, addiction in 8.4% of the cases, and physical disability in 6.1% of the cases. In 4.5% of the cases, other health-related conditions were named. Multiple answers were possible in this question. For a results breakdown by group, see Table 3. No statistically significant differences were found among the groups.

### 3.2. Quantitative Analysis

#### 3.2.1. Primary Outcomes

The mean scale scores for all primary outcomes are presented in Table 4. Participants in the three groups did not differ significantly in their baseline levels of psychological flexibility (AFQ-Y, *F*(2, 210) = 0.14, *p* = 0.87), mindfulness (CAMM, *F*(2, 210) = 0.23, *p* = 0.79), resilience (BRS, *F*(2, 210) = 1.83, *p* = 0.16), mental well-being (WEMWBS, *F*(2, 210) = 2.15, *p* = 0.12), quality of life (KIDSCREEN-10, *F*(2, 210) = 0.49, *p* = 0.61), health complaints (HBSC, *F*(2, 210) = 1.25, *p* = 0.29), and caring-related quality of life (CR-QoL, *F*(2, 201) = 1.92, *p* = 0.15).

In mixed ANOVAs, the time–group interaction effects were nonsignificant for all outcome variables, with the only exception being KIDSCREEN-10 (Table 4). Follow-up analyses indicated a significant main effect of time in the control group, (*F*(2, 104) = 8.38, *p* < 0.001), but not in the ME-WE FTF or ME-WE APP groups, (*F*(2, 73) = 0.20, *p* = 0.82; *F*(2, 30) = 2.61, *p* = 0.09, respectively). The control group increased their scores between T0 and T1, but then decreased them between T1 and T2. All effect sizes (Cohen’s *d*) for differences between T0 and T1 and between T0 and T2 were small.

For both positive and negative outcomes of caring (PANOC), a significant group effect was observed at baseline (PANOC positive, *F*(2, 210) = 6.00, *p* = 0.003; PANOC negative, *F*(2, 210) = 3.40, *p* = 0.04). Post hoc analyses, in which the Bonferroni corrected alpha level of 0.017 was used, indicated that the ME-WE APP group reported significantly lower positive outcomes of caring compared to both the ME-WE FTF (*p* = 0.033, mean difference 95% CI [−4.49, −0.15]) and waitlist groups (*p* = 0.002, mean difference 95% CI [−5.12, −0.97]). Regarding negative outcomes of caring, the post hoc pairwise comparisons were all nonsignificant.

Considering baseline group effects for the positive and negative dimensions of PANOC, we examined the effect of time within each group separately. No statistically significant changes were found in either group for PANOC positive outcomes of caring (Table 4). For PANOC negative outcomes of caring, no statistically significant changes were observed in the ME-WE FTF group, whereas significant changes were detected in the ME-WE APP and waitlist groups over time. Post hoc analyses revealed that in the ME-WE APP group, scores for negative outcomes of caring increased between T0 and T1 (*p* = 0.050, mean difference 95% CI [−2.85, 0.00]) and then decreased between T1 and T2 (*p* = 0.035, mean difference 95% CI [0.01, 3.46]). In the waitlist group, scores for negative outcomes of caring increased between T1 and T2 (*p* = 0.007, mean difference 95% CI [−1.55, −0.19]). However, considering the Bonferroni corrected alpha level of 0.017, only the increase in negative outcomes of caring in the waitlist control group was statistically significant.

#### 3.2.2. Secondary Outcomes

We summed scores on the five ad hoc questions assessing school/training/work attendance and performance. Groups did not differ significantly at the baseline, *F*(2, 210) = 0.63, *p* = 0.54). Furthermore, we did not observe a significant time–group interaction effect, and all observed effect sizes were small (Table 5).

#### 3.2.3. Additional Measures

There were significant between-group differences in caring activities at baseline (MACA total score, *F*(2, 210) = 4.33, *p* = 0.01; MACA personal care, *F*(2, 210) = 5.35, *p* = 0.01; MACA sibling care, *F*(2, 210) = 3.19, *p* = 0.05). Post hoc tests with the Bonferroni adjusted alpha level of 0.017 indicated that participants in the ME-WE APP group had lower MACA total scores compared to the ME-WE FTF group (*p* = 0.020, mean difference 95% CI [−5.79, −0.40]) and the waitlist group (*p* = 0.017, mean difference 95% CI [−5.59, −0.44]). The ME-WE APP group also scored significantly lower on personal care in comparison to the ME-WE FTF group (*p* = 0.004, mean difference 95% CI [−1.76, −0.28]) and the waitlist group (*p* = 0.054, mean difference 95% CI [−1.40, 0.01]), and scored lower on sibling care compared to the ME-WE FTF group (*p* = 0.038, mean difference 95% CI [−1.95, −0.04]). No significant baseline differences were found for domestic tasks (*F*(2, 210) = 1.09, *p* = 0.34), household management (*F*(2, 210) = 1.63, *p* = 0.20), emotional care (*F*(2, 210) = 1.13, *p* = 0.33), or financial/practical care (*F*(2, 210) = 1.85, *p* = 0.16).

For domestic tasks, household management, emotional care, and financial/ practical care, none of the time–group interaction effects were statistically significant (Table 6). For the MACA total score, personal care, and sibling care, we tested time effects separately in each group due to baseline differences. No statistically significant differences were found except for sibling care in the waitlist group (Table 6). Post hoc tests indicated a decrease in scores from T1 to T2 (*p* = 0.021, mean difference 95% CI [0.04, 0.71]).

At baseline, the groups differed in mean levels of satisfaction with overall support (*F*(2, 210) = 7.71, *p* < 0.001). The ME-WE APP group was significantly more satisfied with overall support than both the ME-WE FTF (*p* = 0.030, mean difference 95% CI [0.04, 1.02]) and waitlist groups (*p* < 0.001, mean difference 95% CI [0.31, 1.24]). Due to baseline differences, we examined time effects separately in each group. No statistically significant differences were found (Table 6).

#### 3.2.4. PISA Outcomes

Almost all participants in the intervention groups enjoyed most of the activities: *χ*^2^(1) = 4.72, *p* = 0.03 for the comparison between the ME-WE FTF and ME-WE APP groups at T1, with a higher share in the ME-WE FTF group; *χ*^2^(1) = 1.51, *p* = 0.22 for the comparison between the ME-WE FTF and ME-WE APP groups at T2 (Table 7). Participants expressed strong beliefs that the intervention was worth going to (T1, *χ*^2^(1) = 1.56, *p* = 0.21; T2, *χ*^2^(1) = 3.39, *p* = 0.07) and that it had taught them useful things (T1, *χ*^2^(1) = 1.19, *p* = 0.28; T2, *χ*^2^(1) = 0.56, *p* = 0.45). Most of the participants reported that the intervention made them feel good about themselves (T1, *χ*^2^(1) = 0.21, *p* = 0.65; T2, *χ*^2^(1) = 4.13, *p* = 0.05, with a higher share in the ME-WE FTF group) and about their family (T1, *χ*^2^(1) = 7.84, *p* = 0.01, with a higher share in the ME-WE FTF group; T2, *χ*^2^(1) = 1.84, *p* = 0.18). Approximately half of the participants thought that the person they cared for was better off because they had participated in the intervention (T1, *χ*^2^(1) = 0.45, *p* = 0.50; T2, *χ*^2^(1) = 4.85, *p* = 0.03, with a higher share in the ME-WE FTF group). The impact of the intervention on making new friends was less visible, with no differences between the intervention groups (T1, *χ*^2^(1) = 0.24, *p* = 0.63; T2, *χ*^2^(1) = 0.86, *p* = 0.35) (Table 7).

The results do not suggest major changes in AYCs’ lives compared to the time prior to the intervention (Table 8). Most of the participants in both the ME-WE FTF and ME-WE APP intervention groups replied ‘about the same’ (in comparison to the time before the intervention) to all items on various aspects of their caring role. Approximately a quarter to a third of participants reported that they felt able to choose the level of care they provide more often than before the intervention (T1, *χ*^2^(2) = 1.26, *p* = 0.53; T2, *χ*^2^(2) = 1.96, *p* = 0.38). However, around one fifth of all participants reported that they did caring more often compared to the time before the intervention (T1, *χ*^2^(2) = 1.93, *p* = 0.38; T2, *χ*^2^(2) = 1.78, *p* = 0.41), but also that people were more understanding about their caring tasks than before the intervention (T1, *χ*^2^(2) = 1.51, *p* = 0.47; T2, *χ*^2^(2) = 1.07, *p* = 0.56).

A higher proportion of participants in the ME-WE FTF than in the ME-WE APP intervention group reported that they more often did the caring jobs they disliked (T1, *χ*^2^(2) = 2.16, *p* = 0.34; T2, *χ*^2^(2) = 3.00, *p* = 0.22) and the caring jobs that upset them (T1, *χ*^2^(2) = 1.88, *p* = 0.39; T2, *χ*^2^(2) = 7.97, *p* = 0.02) and worried them the most (T1, *χ*^2^(2) = 5.43, *p* = 0.07; T2, *χ*^2^(2) = 6.65, *p* = 0.04). On the other hand, participants in the ME-WE APP intervention more often reported that other organizations (T1, *χ*^2^(2) = 2.41, *p* = 0.30; T2, *χ*^2^(2) = 2.02, *p* = 0.36) or other people (T1, *χ*^2^(2) = 2.47, *p* = 0.29; T2, *χ*^2^(2) = 4.47; p = 0.11) were providing help for the person they cared for.

A quarter to a third of participants in both groups reported that the intervention had some positive impact on school attendance (T1, *χ*^2^(2) = 2.86, *p* = 0.24; T2, *χ*^2^(2) = 1.19, *p* = 0.55) and performance (T1, *χ*^2^(2) = 0.39, *p* = 0.82; T2, *χ*^2^(2) = 2.11, *p* = 0.35).

### 3.3. Qualitative Analysis

The summary findings of the content analysis of the qualitative data from the AYCs’ evaluation questionnaire across all six countries were synthesized and divided into three main parts, which are described below with the accompanying tables: AYCs’ experiences of caring activities (Table 9), AYCs’ experiences and perceived changes during the COVID-19 pandemic (Table 10), and the Post-Intervention Self-Assessment (PISA). (Table 11). This is followed by a short summary of AYCs’ qualitative feedback regarding the ME-WE app (in the Netherlands and Sweden) (Table 12) and the evaluation survey itself (Table 13).

#### 3.3.1. AYCs’ Experiences of Caring Activities

To gain insights into AYCs’ experiences of caring activities, different aspects were examined such as various caring activities, including both negative and positive experiences. Overall, experiences in the six countries were similar, and some of the most commonly listed caring activities, such as emotional care and domestic tasks, were listed as both the most positive and the most negative experiences. While this may appear contradictory, it also highlights that being a carer is generally perceived as both a positive and a negative experience and may be perceived differently among carers (Table 9).

#### 3.3.2. AYCs’ Experiences and Perceived Changes during the COVID-19 Pandemic

It is important to note that the restrictions following the COVID-19 pandemic differed between countries, which might have affected respondents’ replies to the open-ended questions of the evaluation questionnaire. For instance, some countries faced temporary ‘lock-down’ on one or several occasions, while in Sweden most restrictions focused on recommendations for social distancing and some periods of school closures for upper secondary schools. Thus, the comments differed between the countries. Nevertheless, many of the aspects raised concerned life in general during the pandemic and not specifically the ME-WE intervention.

One of the main differences between AYCs from the different countries is that more respondents from Italy, the Netherlands, and Slovenia reported positive changes rather than negative changes, while the vast majority of AYCs from Sweden and the UK reported negative changes. In Slovenia, online schooling was reported to make life less stressful and reduced the fear of bringing COVID-19 home to their loved ones. From the summaries, both Italy and the Netherlands had AYCs who stated their health and overall well-being had not been affected by the COVID-19 pandemic. Surprisingly, the experiences of AYCs in Sweden were comparatively more negative, even though Sweden opted for less restrictive measures during the COVID-19 pandemic, while in Italy, which was most heavily hit by the COVID-19 pandemic, AYCs reported more positive changes (Table 10).

#### 3.3.3. Post-Intervention Self-Assessment (PISA)

These results should be interpreted by considering that the implementation of interventions differed between the countries due to differing precautionary measures during the COVID-19 pandemic and their duration. In Italy and Slovenia, some AYCs had completed their participation in accordance with the original delivery approach prior to the lockdown. In Slovenia and the Netherlands, COVID-19 interrupted face-to-face and blended intervention delivery, respectively. Meetings were then held online, which meant that in Slovenia and the Netherlands, some AYCs had both face-to face and online ME-WE meetings, while other AYCs only had online meetings. Overall, a greater part of the intervention delivery occurred after the COVID-19 outbreak, meaning that in all six countries, some participants received an intervention that was held completely online (Table 11).

#### 3.3.4. Evaluation of the ME-WE App

The app was used in three countries, the Netherlands, Sweden, and Switzerland, but in the latter, the AYCs who received the intervention did not complete the questionnaires; therefore, data are only available for the Netherlands and Sweden. AYCs in both Sweden and the Netherlands were most satisfied with the appearance of the app and the group exercises. They were most dissatisfied with the usability of the app. Most of the identified problems were technical problems with the app and network issues. Finally, most of the respondents in both Sweden and the Netherlands stated that the app provided functionalities that they could not find in other apps/services (Table 12).

#### 3.3.5. AYCs’ Feedback on the Evaluation Questionnaire

For some AYCs, the questionnaire aroused emotions that were perceived as both positive and negative. Negative comments related to the length and content of the survey. A few AYCs found some questions confusing or difficult to understand or needed help to complete the survey. Positive comments focused on the questions acting as a form of self-reflection over their situation (Table 13).

## 4. Discussion

Our research examined whether and to what extent the ME-WE intervention promoted favorable changes in AYCs’ mental health and well-being outcomes compared to the waitlist group. Based on the quantitative analyses of the evaluation questionnaires for AYCs, it is challenging to make conclusive claims about whether and how the ME-WE model increased participant AYCs’ psychological flexibility, mindfulness, resilience, mental well-being, and quality of life. Although participants in both intervention groups (ME-WE FTF and ME-WE APP) seemed to slightly increase their resilience and mindfulness skills, differences with the waitlist control group were generally low and nonsignificant at a statistical level. A possible explanation for this is the fact that most of the control groups received some form of basic support (instead of no support, as usually happens in some/most countries), which was possibly also sufficient to stimulate positive outcomes in the control groups. This may have reduced the difference between the intervention and waitlist control groups. It also seems that both intervention groups improved their mental well-being, but also perceived a lower general health-related quality of life over time while they retained similar levels of health complaints. The results regarding changes in psychological flexibility are inconclusive.

The intervention might not have had a long-term effect on positive outcomes of caring, but there seems to have been a positive effect on decreasing negative outcomes of caring. The qualitative analysis somehow also shows a more encouraging outlook on the positive outcomes of caring as well.

The results do not suggest an impact of the intervention on school attendance or performance, although the Post-Intervention Self-Assessment (PISA) indicated that the intervention could have had some positive impact on school attendance and performance. However, these results should be interpreted with caution, as most of the intervention activities took place during the COVID-19 pandemic. Hence, the possible impact of the pandemic on school performance should be taken into consideration [37]. Nevertheless, there is a need for more long-term intervention or follow-up support. This is a relevant step to take as caring responsibilities have generally been shown to have a negative impact on school attendance [38,39], which in turn can negatively affect AYCs’ future lives [8,9,10].

Further, the abovementioned outcomes also need to take into consideration the possible impact of COVID-19 on AYCs’ quality of life. Overall, the effect of the pandemic on the quantitative results is not entirely clear. The results of the qualitative analysis suggest that the pandemic had a negative impact on AYCs’ lives, caring burden, health, and school results, and some of them perceived that they did not receive the support they needed during the pandemic. Among the negative aspects, the most frequently mentioned was social distancing from friends, which in some cases led to depression and anxiety. The negative impact of the COVID-19 pandemic on young carers has also been reported in other studies (e.g., [40,41]). However, in the present study there were some AYCs who described a positive impact of the pandemic on their lives. This can be especially observed in Italy and Slovenia, where AYCs reported having more time for themselves, being more relaxed, and spending more time with their families.

The findings from the Post-Intervention Self-Assessment (PISA) are arguably the most encouraging takeaway from the study in the six countries. Based on the statistical analyses of the primary outcomes as described above, it was not possible to fully confirm the effectiveness of the ME-WE intervention. However, the ME-WE intervention was well received by participating AYCs. Participants generally enjoyed most of the activities and believed that the intervention was worth going to. The ME-WE intervention helped participants to deal with stressful thoughts and feelings, to know more about themselves, to be kind to themselves, to find meaning, energy, and power, to feel good and relaxed, and not to feel alone by sharing experiences. These results may be an effect of the ME-WE model’s focus on personal values, flexible self-view, and future opportunities [1,14]. Somehow less prominent was the impact of the intervention on making new friends. One of the reasons could be that participants already knew each other in some of the groups (due to, for example, attending the same school). Also, the online delivery of the intervention could possibly have hindered the opportunity to make new friends, yet at the same time, several AYCs reported that the ME-WE app and video-conferencing system afforded them a greater degree of anonymity, which they personally preferred. However, in terms of the findings regarding the comparison of how AYCs in the ME-WE APP and ME-WE FTF groups experienced the intervention, there were no major differences highlighted.

There were, however, some negative comments from participants regarding the intervention, especially concerning the length of sessions and the home assignments. Further, several AYCs commented that the exercises and group discussions lacked a sufficient focus on the caring role. Thus, the ME-WE model needs some further refinements in order to fully meet AYCs’ individual needs and preferences.

Overall, participants in the ME-WE APP perceived more support in their caring role in comparison to the ME-WE FTF and waitlist participants, but both intervention groups perceived less support at the post-intervention assessment. One possible explanation for this could be related to the intervention enhancing AYCs’ acknowledgment of their caring role and consequently more critically evaluating the support they are currently receiving. As most of the data were collected during the COVID-19 pandemic, it is likely that this result also reflects the impact of the pandemic, especially in terms of limited access to existing support services and facilities, especially during the first wave of the pandemic [42]. The results also suggest that AYCs in the ME-WE APP group experienced that they received more support in their caring role due to the intervention, which AYCs in ME-WE FTF group did not. One possible explanation could be that AYCs in the ME-WE APP group had access to the ME-WE young carers’ mobile app. As the app offers professional and peer support both during, in between and after the intervention sessions, for instance via the information pages and the chat, it is possible that they felt more support because of this facility. However, this could also reflect country-specific contexts.

To summarize, it can be argued that the intervention might have contributed to positive outcomes of caring as well as the mental well-being of AYCs, despite the generally nonsignificant statistical results. Importantly, participants reported enjoying the activities and that the intervention was worth going to. Some of the participants recognized positive aspects of caring and they more often felt able to opt out of uncomfortable and upsetting caring activities, compared to prior to the intervention. The results indicate that the ME-WE model has the potential of contributing to an increased well-being and to boosting AYCs’ resilience, as well as to learning ways to cope with stressful events and negative emotions. It could give AYCs the confidence, energy, and tools they need to influence their situation in a positive direction, for example by seeking help or support from an adult. However, to find out if the model mitigates the negative impact of psychosocial and environmental factors, a long-term study is needed with larger sample sizes.

### Limitations

The main limitations of the present study include the recruitment difficulties, screening failures, and high levels of dropouts, both before (e.g., in Sweden and Switzerland) and during the intervention (e.g., in Slovenia and the UK). The motivation of AYCs was probably one of the main challenges, as young people overall showed great interest in the project, but very few registered their interest in participating in the study. Sometimes this was due to pragmatic reasons of having limited time to participate in the study, such as challenges of combining schoolwork, traveling home after school hours, having responsibilities at home, and sometimes performing paid work or participating in team sports. Moreover, due to a prevailing rather under-established concept of young carers, many young carers do not recognize themselves as ‘caregivers’ and find it a normal part of life to care for their loved ones [43,44,45]. Hence, it is important to take into consideration various cultural aspects, i.e., ‘family culture’, that likely entail that caring tasks and the resulting burdens should remain confidential [44]. Furthermore, AYCs might not be willing to deal with such a difficult and sensitive topic in public, which can trigger a range of emotions as well as the risk of stigmatization [45]. In recognizing the various challenges and efforts undertaken, it is important to acknowledge as a limitation the substantially lower number of study participants than initially anticipated, which might affect the robustness of statistical analysis. The total final sample size included 107 AYCs in the intervention groups (all combined) and 106 AYCs in the waitlist control group, a figure almost halved from the target sample size (263 AYCs in the intervention groups and 263 AYCs in the control group). Recruitment was low and participant engagement was not fully realized despite the significant recruitment efforts. High dropout levels (particularly in Slovenia and the UK) indicate difficulties in maintaining participant involvement in the study. The requirement to fill in online surveys containing a relatively comprehensive set of sensitive questions might have discouraged some AYCs from participating. Certain measures used in the research (i.e., AFQ, CAMM, and BRS) had not been previously validated with an AYC population. However, all the existing tools selected for this study were carefully chosen based on their validation for adolescent samples. Where necessary, these measures were translated following a rigorous procedure in each participating country. Unfortunately, the country-specific sample sizes did not allow for the robust testing of structural validity through factor analysis. Nonetheless, we computed reliability coefficients for each measure across the different countries, and these coefficients were found to be acceptable (i.e., ≥0.60 [46]) across all contexts, supporting the internal consistency of the scales used. Given the validated nature of these measures and the satisfactory reliability observed, we consider the scores used in this study to be reliable.

The ME-WE RCT study was carried out during highly challenging circumstances, including COVID-19 social distancing measures and intermittent lockdowns across all participating countries. The consequences of the pandemic led to the complete halt of recruitment activities in the six countries, prompting the adoption of social-media-driven individual recruitment campaigns in Switzerland and Sweden. The delivery of the intervention itself was severely disrupted by the onset of the pandemic, with the Netherlands and Slovenia experiencing the initial lockdown phase that impacted Europe. Coupled with the small sample sizes, the COVID-19 period posed an additional set of changes (both in terms of social environment that differed across countries as well as in individual challenges that AYCs met during this time), thus resulting in a statistical analysis that could not encompass all relevant factors at the same time. All participants in both intervention groups (including dropouts and compassionate cases) were included in the qualitative content analysis of the open-ended items, thus making qualitative analysis richer and more informative. In some cases, the control group participants did not attend a meeting, but they filled in the evaluation questionnaire.

Geographical limitations of the study should also be acknowledged. The UK sample only included participants in England, while AYCs in the other three nations of the UK (Scotland, Wales, and Northern Ireland) were not included in the study. Similarly, in Switzerland, participants were only recruited from the German-speaking areas (and not from Italian- or French-speaking parts). Thus, the results should be carefully considered before immediate extrapolation to the entirety of the UK or Switzerland. Furthermore, any generalization of the results beyond participating countries is currently questionable and requires further research.

In certain cases, deviations from the study protocol were noted. The time intervals between all three assessment points did not always follow the protocol due to several occurring issues and hence the period between T1 and T2 varied from 1.5 weeks to over three months. This should be borne in mind when making comparisons between the intervention and control groups. The three-month follow-up might also be too short to consider findings as long-term effects, and further studies are needed to confirm the long- lasting effects of the intervention. Owing to smaller sample sizes, data were assessed for participants that attended at least 50% of the sessions.

There were some instruments or ad hoc measures that have been excluded from presentation in this paper owing to challenges with open-ended response categories. These include the Brief Social Support Questionnaire [47], consisting of six items with the number of support sources as the response option, and two open-ended ad hoc questions on the overall amount of caring, with the number of hours (hours of caring per week for a typical day during the week and the weekend) as the response option. We deemed the answers in many cases not to be entirely reliable (e.g., estimated hours of caring were higher than the number of hours in a day, and answers to the number of people AYCs could count on were often non-numerical, such as ‘many’ or ‘a lot’), hence preventing us from computing reliable descriptive statistics.

The country-specific measures, such as ad hoc questions assessing formal support or services that AYCs and someone in their family received (i.e., care package, equipment, transportation assistance, benefits, and allowances), and whether school staff, other family members, and friends had been trusted or knew about their caring situation, were omitted from analysis presented in this manuscript.

Furthermore, in this paper, we only presented answers to open-ended questions on the app evaluation. However, the app evaluation included additional question on users’ satisfaction with the app (ranging from 0 ‘Totally dissatisfied’ to 10 ‘Totally satisfied’), two multiple-answer questions on which of the app features AYCs were most satisfied or dissatisfied with (e.g., appearance, content), as well as several questions on whether and to what extent the app had been useful in AYCs’ life and whether they would recommend it to other AYCs. Due to the small number of collected answers, the research team opted for omitting this part of the analysis from this paper.

Owing to the very high risk of dropout of recruited AYCs experienced during the pandemic in Sweden (recruited AYCs wanted to commence the ME-WE group immediately), the double-blind assignment of participants to trial groups was omitted, and due to the lack of resources and impact of the pandemic, the intervention was solely offered to participants in Swedish. It is important to note that all Swedish data were kept for analysis to boost the sample size. The recruitment and fieldwork proved to be especially challenging in Switzerland as only one fully evaluated participant was achieved. The case was omitted from analysis for the purpose of not disclosing personal data.

Furthermore, emotional distress is arguably heightened during a pandemic, which might have mitigated the impact of the intervention on AYCs’ overall health and mental well-being [48]. At this point, it was not feasible to clearly distinguish the possible impact of COVID-19, especially with regards to the quantitative analysis. Hence, further analysis in this regard, as well as repeated intervention runs to increase sample sizes, should be encouraged if further funding opportunities become available.

In general, any comparisons deducted from this study should be interpreted with caution. There was uneven recruitment to the intervention and control group, making the statistical comparisons even more challenging, especially at the national level. Although on an aggregate cross-national level a more balanced sample size was achieved, the research team was prevented from making straightforward comparisons. The reason for this lies not only in the challenges to ensure sufficient sample sizes, but also, equally importantly, with regards to having a versatile set of recruitment strategies and delivery approaches to be considered across the six countries. Clearly, such a consideration is highly limited with the small sample sizes achieved.

## 5. Conclusions

The ME-WE model distinguishes itself as the first-ever RCT involving AYCs. Its innovative inclusion of six European countries, each with varying levels of approaches and awareness of young carers, marks a significant milestone at the European level. Furthermore, this study represents the first application of the DNA-V program to a young carers’ population. Despite several methodological limitations and caution required for data interpretation, the results indicate some positive impacts. Despite the numerous challenges faced by the project, the ME-WE model qualitatively benefitted the mental health and overall well-being of the AYCs, as evidenced by their own first-hand accounts.

A substantial portion of the recruitment and the intervention delivery occurred during the COVID-19 pandemic, making the results uniquely reflective of this particular period. The pandemic posed various challenges, but the need to redesign the ME-WE intervention delivery from face-to-face to entirely online created an unexpected opportunity. This shift effectively served as a pilot study for the online delivery mode.

The online version of the ME-WE intervention provides a novel supportive program for AYCs residing in hard-to-reach areas, regions lacking adequate support, or those not enrolled in school or education. Consequently, these AYCs can access the ME-WE intervention due to its online format. This transition to online delivery ensures greater access to the ME-WE program, enabling AYCs, who are often more hidden than their peers, to benefit from a validated, supportive program.

## Figures and Tables

**Figure 1 healthcare-12-02124-f001:**
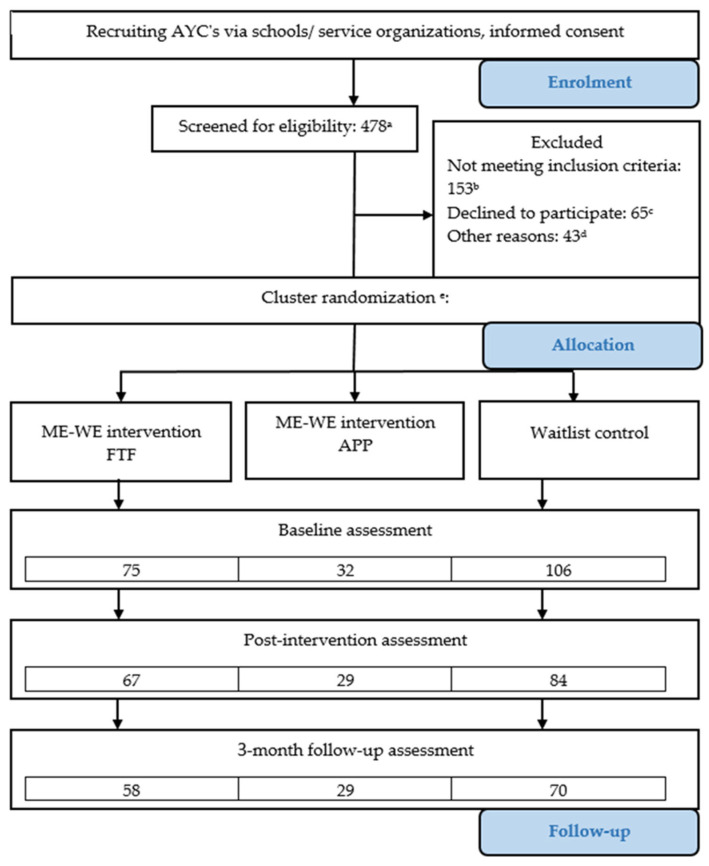
ME-WE research flow diagram. ^a^ Total number of young persons who applied to the study and were screened, including participants who consented to participate. ^b^ Several young people who did not meet the screening criteria (including compassionate cases). Mostly they were outside the age range. Those on psychotropic medication, and those either in current receipt of a psychotherapeutic intervention or mental health counseling or planning to receive such therapies during the ME-WE RCT study were also deemed compassionate cases in the UK and Sweden. ^c^ Total number of AYCs eligible for participation (applied and were screened positively) but who eventually did not start the intervention. ^d^ Number of participants who were excluded from the analysis due to attending less than 50% of sessions. ^e^ Swiss participants (*n* = 4) are included in the figures until ‘cluster randomization’ and excluded from the assessment figures.

**Table 1 healthcare-12-02124-t001:** Sociodemographic characteristics of both intervention groups and waitlist control group with corresponding test statistics.

Characteristic	ME-WE FTF (*n* = 75)	ME-WE APP (*n* = 32)	Waitlist (*n* = 106)	Test Statistics
Age, *M* (*SD*)	16.27 (0.83)	16.25 (0.74)	16.47 (0.92)	*F*(2, 210) = 1.46, *p* = 0.23
Gender, %				*χ*^2^ = 5.76, *p* = 0.19 *
Male	22.7	12.5	15.2	
Female	74.7	78.1	82.9	
Other/Undisclosed	0	6.3	1.0	
Area of living, %				*χ*^2^ = 5.03, *p* = 0.76 *
A big city	14.7	9.4	18.3	
The suburbs or outskirts of a big city	12.0	18.8	11.5	
A town or small city	44.0	50.0	44.2	
A country village	21.3	18.8	23.1	
A farm or home in the countryside	8.0	3.1	2.9	
Country of birth, %				*χ*^2^ = 0.53, *p* = 0.80 *
In the country of living	89.2	93.8	91.4	
In another country	10.8	6.3	8.6	
Country of mother’s birth, %				*χ*^2^ = 1.65, *p* = 0.45 *
In the country of living	80.0	87.5	86.7	
In another country	20.0	12.5	13.3	
Country of father’s birth, %				*χ*^2^(2) = 1.27, *p* = 0.53
In the country of living	82.4	90.6	82.9	
In another country	17.6	9.4	17.1	
Living with, %				
Mother	95.9	84.4	97.1	*χ*^2^ = 6.61, *p* = 0.03 *
Father	74.3	62.5	74.3	*χ*^2^(2) = 1.90, *p* = 0.39
Stepmother (or parent’s girlfriend)	0	3.1	3.8	*χ*^2^ = 2.94, *p* = 0.23 *
Stepfather (or parent’s boyfriend)	5.4	6.3	5.7	*χ*^2^ = 0.21, *p* = 1.00 *
Brother(s)	43.2	37.5	51.4	*χ*^2^(2) = 2.38, *p* = 0.30
Sister(s)	55.4	37.5	52.4	*χ*^2^(2) = 2.99, *p* = 0.23
Grandmother	20.3	3.1	13.3	*χ*^2^ = 5.60, *p* = 0.06 *
Grandfather	12.2	3.1	5.7	*χ*^2^ = 3.13, *p* = 0.23 *
In a foster home	0	6.3	1.0	*χ*^2^ = 4.57, *p* = 0.06 *
In children’s home	0	3.1	1.0	*χ*^2^ = 2.36, *p* = 0.40 *
Someone or somewhere else	5.4	12.5	1.9	*χ*^2^ = 5.74, *p* = 0.04 *

* Fisher’s exact test.

**Table 2 healthcare-12-02124-t002:** Characteristics of participants’ family members in need of help or support.

Characteristic	ME-WE FTF (*n* = 70)	ME-WE APP (*n* = 47)	Waitlist (*n* = 125)	Test Statistics
Family member, %				
Mother	30.0	23.4	27.0	*χ*^2^(2) = 0.62, *p* = 0.73
Father	14.3	14.9	7.1	*χ*^2^(2) = 3.48, *p* = 0.18
Stepmother (or parent’s girlfriend)	0	0	0	N/A
Stepfather (or parent’s boyfriend)	0	0	0.8	*χ*^2^ = 1.19, *p* = 1.00 *
Brother	11.4	23.4	9.5	*χ*^2^(2) = 6.08, *p* = 0.05
Sister	17.1	23.4	11.9	*χ*^2^(2) = 3.60, *p* = 0.17
Grandmother	12.9	4.3	18.3	*χ*^2^(2) = 5.68, *p* = 0.06
Grandfather	4.3	2.1	9.5	*χ*^2^ = 3.35, *p* = 0.19 *
Aunt	1.4	2.1	3.2	*χ*^2^ = 0.53, *p* = 0.86 *
Uncle	2.9	0	4.0	*χ*^2^ = 1.51, *p* = 0.52 *
Cousin	0	0	5.6	*χ*^2^ = 5.40, *p* = 0.04 *
Other person	5.7	6.4	3.2	*χ*^2^ = 1.47, *p* = 0.54 *
Live with this person, %	76.5	58.3	67.5	*χ*^2^(2) = 4.33, *p* = 0.12
Type of health-related condition, %				
Physical illness	31.9	15.9	35.7	*χ*^2^(2) = 8.55, *p* = 0.01
Physical disability	20.2	14.3	20.2	*χ*^2^(2) = 1.16, *p* = 0.56
Ill mental health	20.2	28.6	16.1	*χ*^2^(2) = 4.55, *p* = 0.10
Cognitive impairments	13.8	31.7	20.8	*χ*^2^(2) = 7.32, *p* = 0.03
Addiction	4.3	1.6	2.4	*χ*^2^ = 1.10, *p* = 0.66 *
Other health-related conditions	9.6	7.9	4.8	*χ*^2^ = 2.56, *p* = 0.27 *
Age, *M* (*SD,* range)	40.66 (22.58, 3–86)	36.62 (20.99, 7–85)	46.04 (24.80, 3–90)	*F*(2, 239) = 3.11, *p* = 0.05

* Fisher’s exact test.

**Table 3 healthcare-12-02124-t003:** Characteristics of participants’ close friends in need of help or support.

Characteristic	ME-WE FTF (*n* = 32)	ME-WE APP (*n* = 30)	Waitlist (*n* = 83)	Test Statistics
Close friend, %				
Girlfriend/Boyfriend	6.1	20.0	4.8	*χ*^2^ = 5.89, *p* = 0.05 *
Friend	69.7	73.3	78.3	*χ*^2^(2) = 1.03, *p* = 0.58
Colleague	12.1	0	8.4	*χ*^2^ = 3.65, *p* = 0.16 *
Neighbor	3.0	0	4.8	*χ*^2^ = 1.09, *p* = 0.82 *
Ex-girlfriend/Ex-boyfriend	0	3.3	2.4	*χ*^2^ = 1.01, *p* = 0.78 *
Cohabitant/Roommate	0	0	0	-
Other person	9.1	3.3	1.2	*χ*^2^ = 4.08, *p* = 0.07 *
Live with this person, %	12.1	0	1.2	*χ*^2^ = 6.75, *p* = 0.03 *
Type of health-related condition, %				
Physical illness	9.8	2.6	16.0	*χ*^2^ = 4.95, *p* = 0.09 *
Physical disability	4.9	10.5	5.0	*χ*^2^ = 1.64, *p* = 0.40 *
I’ll mental health	58.5	60.5	56.0	*χ*^2^(2) = 0.25, *p* = 0.88
Cognitive impairments	12.2	15.8	10.0	*χ*^2^ = 1.08, *p* = 0.57 *
Addiction	9.8	7.9	8.0	*χ*^2^ = 0.28, *p* = 0.93 *
Other health-related conditions	4.9	2.6	5.0	*χ*^2^ = 0.37, *p* = 1.00 *
Age, *M* (*SD*, range)	18.28 (8.29, 14–50)	17.00 (2.42, 14–24)	19.76 (13.23, 9–74)	*F*(2, 142) = 0.78, *p* = 0.46

ME-WE FTF = face-to-face ME-WE intervention; ME-WE APP = app-based ME-WE intervention. * Fisher’s exact test.

**Table 4 healthcare-12-02124-t004:** Mean scores (*SD*) of primary outcomes at all three evaluation points, and effect size (Cohen’s *d*) for differences between baseline (T0) and post-intervention (T1) and between baseline (T0) and follow-up (T2) for ME-WE FTF (*n* = 75), ME-WE APP (*n* = 32), and waitlist control group (*n* = 106).

	Mean (*SD*)			Cohen’s *d*		Effect
	T0	T1	T2	T0-T1	T0-T2	Time–Group
AFQ-Y						Wilks’ *λ* = 0.98, *F*(4, 418) = 1.18, *p* = 0.32, *η*^2^ = 0.011
ME-WE FTF	12.65 (6.34)	12.69 (5.63)	12.08 (5.40)	0.01	0.10	
ME-WE APP	12.71 (6.72)	14.03 (6.69)	12.88 (6.07)	0.20	0.03	
Waitlist	12.20 (6.30)	12.96 (5.34)	13.09 (4.74)	0.13	0.16	
CAMM						Wilks’ *λ* = 0.99, *F*(4, 418) = 0.37, *p* = 0.83, *η*^2^ = 0.004
ME-WE FTF	20.44 (7.56)	20.50 (6.40)	21.09 (6.69)	0.01	0.09	
ME-WE APP	19.50 (6.77)	18.83 (7.27)	20.41 (7.23)	0.09	0.13	
Waitlist	20.23 (5.67)	19.79 (4.86)	20.42 (4.74)	0.08	0.04	
BRS						Wilks’ *λ* = 0.99, *F*(4, 418) = 0.41, *p* = 0.80, *η*^2^ = 0.004
ME-WE FTF	17.74 (3.43)	18.59 (3.89)	18.46 (3.98)	0.23	0.19	
ME-WE APP	16.38 (4.10)	16.47 (4.34)	16.92 (4.24)	0.02	0.13	
Waitlist	16.93 (3.68)	17.44 (3.26)	17.69 (3.36)	0.14	0.21	
WEMWBS						Wilks’ *λ* = 0.98, *F*(4, 418) = 1.22, *p* = 0.30, *η*^2^ = 0.012
ME-WE FTF	45.97 (9.43)	48.37 (7.85)	48.11(6.78)	0.28	0.26	
ME-WE APP	44.81 (6.76)	46.14 (9.24)	46.64(10.70)	0.16	0.20	
Waitlist	48.03 (8.96)	48.26 (8.40)	47.90 (7.74)	0.03	0.02	
KIDSCREEN-10						Wilks’ *λ* = 0.95, *F*(4, 418) = 2.74, *p* = 0.03, *η*^2^ = 0.026
ME-WE FTF	33.15 (6.15)	33.34 (5.59)	32.92 (5.68)	0.03	0.04	
ME-WE APP	34.15 (6.07)	32.27 (6.73)	32.29 (6.86)	0.29	0.29	
Waitlist	34.02 (6.53)	34.99 (6.38)	32.82 (5.64)	0.15	0.20	
HBSC						Wilks’ *λ* = 0.96, *F*(4, 418) = 1.97, *p* = 0.10, *η*^2^ = 0.019
ME-WE FTF	17.34 (6.88)	17.44 (6.17)	18.19 (6.74)	0.02	0.12	
ME-WE APP	19.58 (5.69)	21.30 (6.92)	20.14 (6.41)	0.27	0.09	
Waitlist	18.36 (7.17)	17.73 (6.03)	17.83 (5.21)	0.10	0.08	
PANOC-P ^a^						
ME-WE FTF	13.26 (4.02)	13.14 (4.37)	13.16 (3.75)	0.03	0.02	Wilks’ *λ* = 1.00, *F*(2, 73) = 0.34, *p* = 0.97, *η*^2^ = 0.001
ME-WE APP	10.94 (5.16)	11.61 (5.21)	10.85 (4.36)	0.13	0.02	Wilks’ *λ* = 0.96, *F*(2, 30) = 0.63, *p* = 0.54, *η*^2^ = 0.040
Waitlist	13.98 (4.34)	13.61 (4.26)	13.93 (3.80)	0.09	0.01	Wilks’ *λ* = 0.99, *F*(2, 104) = 0.79, *p* = 0.46, *η*^2^ = 0.015
PANOC-N ^a^						
ME-WE FTF	4.79 (3.87)	4.03 (3.25)	4.16 (3.45)	0.21	0.17	Wilks’ *λ* = 0.96, *F*(2, 73) = 1.47, *p* = 0.24, *η*^2^ = 0.039
ME-WE APP	5.52 (4.41)	6.95 (5.43)	5.17 (4.46)	0.29	0.08	Wilks’ *λ* = 0.78, *F*(2, 30) = 4.36, *p* = 0.02, *η*^2^ = 0.225
Waitlist	3.70 (3.71)	3.37 (3.43)	4.25 (3.30)	0.09	0.15	Wilks’ *λ* = 0.91, *F*(2, 104) = 4.87, *p* = 0.01, *η*^2^ = 0.089
CR-QoL						
ME-WE FTF	0.33 (0.51)	0.21 (0.40)	0.33 (0.50)	0.26	0	
ME-WE APP	0. 47 (0.71)	0.59 (0.70)	0.48 (0.74)	0.17	0.01	Wilks’ *λ* = 0.96, *F*(4, 418) = 1.93, *p* = 0.11, *η*^2^ = 0.018
Waitlist	0.25 (0.53)	0.22 (0.42)	0.26 (0.36)	0.06	0.02	

AFQ-Y = Avoidance and Fusion Questionnaire for Youth; CAMM = Child and Adolescent Mindfulness Measure; BRS = Brief Resilience Scale; WEMWBS = Warwick Edinburgh Mental Well-Being Scale; KIDSCREEN-10; HBSC Symptom Checklist; PANOC-P = Positive and Negative Outcomes of Caring—Positive outcome subscale; PANOC-N = Positive and Negative Outcomes of Caring—Negative outcome subscale; CR-QoL = caring-related quality of life. ME-WE FTF = face-to-face ME-WE intervention; ME-WE APP = app-based ME-WE intervention. ^a^ Due to baseline differences, time effects were tested separately in each group.

**Table 5 healthcare-12-02124-t005:** Mean scores (*SD*) of secondary outcomes at all three evaluation points, and effect size (Cohen’s *d*) for differences between baseline (T0) and post-intervention (T1) and between baseline (T0) and follow-up (T2) for ME-WE FTF (*n* = 75), ME-WE APP (*n* = 32), and waitlist control group (*n* = 106).

		Mean (*SD*)		Cohen’s *d*		Effect
Group	T0	T1	T2	T0-T1	T0-T2	Time–Group
ME-WE FTF	9.55 (3.21)	8.68 (3.00)	8.19 (2.42)	0.28	0.48	Wilks’ *λ* = 0.96, *F*(4, 418) = 1.95, *p* = 0.10, *η*^2^ = 0.018
ME-WE APP	9.88 (3.00)	10.14 (3.63)	9.09 (3.30)	0.08	0.25	
Waitlist	9.21 (3.25)	9.13 (3.24)	8.93 (3.11)	0.02	0.09	

**Table 6 healthcare-12-02124-t006:** Mean scores (*SD*) of additional measures at all three evaluation points, and effect size (Cohen’s *d*) for differences between baseline (T0) and post-intervention (T1) and between baseline (T0) and follow-up (T2) for ME-WE FTF (*n* = 75), ME-WE APP (*n* = 32), and waitlist control group (*n* = 106).

		Mean (*SD*)		Cohen’s *d*		Effect
	T0	T1	T2	T0-T1	T0-T2	Time–Group
MACA total ^a^						
ME-WE FTF	14.60 (5.53)	13.57 (5.21)	13.89 (5.21)	0.19	0.13	Wilks’ *λ* = 0.94, *F*(2, 73) = 2.32, *p* = 0.11, *η*^2^ = 0.060
ME-WE APP	11.50 (4.21)	11.97 (4.93)	11.04 (4.48)	0.10	0.11	Wilks’ *λ* = 0.94, *F*(2, 30) = 0.95, *p* = 0.40, *η*^2^ = 0.059
Waitlist	14.51 (5.63)	14.76 (5.34)	13.75 (4.37)	0.05	0.15	Wilks’ *λ* = 0.96, *F*(2, 104) = 2.47, *p* = 0.09, *η*^2^ = 0.045
Domestic tasks						Wilks’ *λ* = 0.98, *F*(4, 418) = 1.25, *p* = 0.29, *η*^2^ = 0.012
ME-WE FTF	4.22 (1.41)	4.10 (1.40)	4.26 (1.36)	0.09	0.03	
ME-WE APP	3.88 (1.36)	3.99 (1.40)	3.64 (1.46)	0.08	0.17	
Waitlist	4.29 (1.43)	4.28 (1.34)	4.14 (1.14)	0.01	0.12	
Household tasks						Wilks’ *λ* = 0.96, *F*(4, 418) = 2.05, *p* = 0.09, *η*^2^ = 0.019
ME-WE FTF	3.07 (1.20)	2.86 (1.41)	3.17 (1.41)	0.16	0.08	
ME-WE APP	2.88 (1.66)	2.98 (1.36)	2.70 (1.58)	0.07	0.11	
Waitlist	3.33 (1.43)	3.43 (1.28)	3.28 (1.04)	0.07	0.04	
Personal care ^a^						
ME-WE FTF	1.21 (1.75)	1.15 (1.54)	1.10 (1.52)	0.03	0.07	Wilks’ *λ* = 0.99, *F*(2, 73) = 0.39, *p* = 0.68, *η*^2^ = 0.011
ME-WE APP	0.19 (0.64)	0.31 (0.77)	0.30 (0.68)	0.17	0.17	Wilks’ *λ* = 0.96, *F*(2, 30) = 0.57, *p* = 0.57, *η*^2^ = 0.037
Waitlist	0.88 (1.45)	0.96 (1.34)	0.83 (1.07)	0.05	0.04	Wilks’ *λ* = 0.99, *F*(2, 104) = 0.64, *p* = 0.53, *η*^2^ = 0.012
Emotional care						Wilks’ *λ* = 0.99, *F*(4, 418) = 0.30, *p* = 0.88, *η*^2^ = 0.003
ME-WE FTF	3.03 (1.76)	2.91 (1.61)	2.73 (1.51)	0.07	0.18	
ME-WE APP	2.81 (1.35)	2.88 (1.64)	2.87 (2.03)	0.04	0.03	
Waitlist	3.29 (1.77)	3.14 (1.54)	3.09 (1.36)	0.09	0.13	
Sibling care ^a^						
ME-WE FTF	2.25 (1.95)	1.96 (1.65)	1.85 (1.65)	0.16	0.22	Wilks’ *λ* = 0.94, *F*(2, 73) = 2.19, *p* = 0.12, *η*^2^ = 0.057
ME-WE APP	1.25 (1.67)	1.27 (1.85)	1.25 (1.60)	0.01	0.00	Wilks’ *λ* = 1.00, *F*(2, 30) = 0.01, *p* = 0.99, *η*^2^ = 0.001
Waitlist	1.81 (1.95)	1.97 (1.69)	1.59 (1.43)	0.09	0.13	Wilks’ *λ* = 0.93, *F*(2, 104) = 3.74, *p* = 0.03, *η*^2^ = 0.067
Financial care						Wilks’ *λ* = 0.96, *F*(4, 418) = 2.08, *p* = 0.08, *η*^2^ = 0.020
ME-WE FTF	0.82 (1.12)	0.59 (0.76)	0.78 (0.83)	0.24	0.04	
ME-WE APP	0.50 (0.88)	0.55 (0.95)	0.29 (0.51)	0.06	0.30	
Waitlist	0.91 (1.04)	0.99 (1.21)	0.82 (0.97)	0.07	0.08	
Overall support ^a^						
ME-WE FTF	2.19 (1.05)	2.09 (0.97)	2.18 (0.99)	0.10	0.01	Wilks’ *λ* = 0.99, *F*(2, 73) = 0.35, *p* = 0.71, *η*^2^ = 0.009
ME-WE APP	2.72 (1.22)	2.51 (0.94)	2.52 (1.24)	0.19	0.16	Wilks’ *λ* = 0.96, *F*(2, 30) = 0.66, *p* = 0.52, *η*^2^ = 0.042
Waitlist	1.95 (0.84)	2.02 (0.93)	2.05 (0.63)	0.08	0.14	Wilks’ *λ* = 0.99, *F*(2, 104) = 0.64, *p* = 0.53, *η*^2^ = 0.012

MACA = Multidimensional Assessment of Caring Activities; ME-WE FTF = face-to-face ME-WE intervention; ME-WE APP = app-based ME-WE intervention. ^a^ Due to baseline differences, time effects were tested separately in each group.

**Table 7 healthcare-12-02124-t007:** Proportion of participants who responded ‘Yes’ to PISA evaluation questions for the ME-WE FTF (*n* = 67 at T1, *n* = 58 at T2) and ME-WE APP (*n* = 29 at T1 and T2) intervention groups.

PISA Item	Time	ME-WE FTF	ME-WE APP
I enjoyed most of the activities.	T1	100	93.1
	T2	98.2	92.9
The intervention helped me make new friends.	T1	56.7	62.1
	T2	50.0	60.7
The intervention taught me useful things.	T1	95.5	89.7
	T2	87.5	92.9
The intervention was worth going to.	T1	93.9	86.2
	T2	94.6	82.1
The intervention made me feel good about myself.	T1	86.4	82.8
	T2	89.1	71.4
The intervention made me feel good about my family.	T1	77.3	48.3
	T2	78.2	64.3
The person I care for is better off because I participated in the intervention.	T1	52.3	44.8
	T2	69.8	44.4

ME-WE FTF = face-to-face ME-WE intervention; ME-WE APP = app-based ME-WE intervention.

**Table 8 healthcare-12-02124-t008:** Proportion of participants who responded ‘More often than before the intervention’ to PISA evaluation questions for the ME-WE FTF (*n* = 67 at T1, *n* = 58 at T2) and ME-WE APP (*n* = 29 at T1 and T2) intervention groups.

PISA Item	Time	ME-WE FTF	ME-WE APP
I feel able to choose the level of care I provide.	T1	23.4	34.5
	T2	28.8	25.0
I do caring.	T1	20.3	20.7
	T2	15.1	21.4
I do the caring jobs that I dislike.	T1	17.2	6.9
	T2	11.3	3.6
I do the caring jobs that upset me.	T1	9.4	3.4
	T2	17.0	0.0
I do the caring jobs that worry me the most.	T1	18.8	3.4
	T2	19.2	0.0
Other organizations are providing help for the person I care for.	T1	8.2	13.8
	T2	3.9	7.4
Other people are providing help for the person I care for.	T1	15.6	20.7
	T2	11.5	22.2
People are understanding of the caring jobs that I do.	T1	21.3	24.1
	T2	25.0	22.2
I attend school or college.	T1	18.2	24.1
	T2	16.7	14.3
I do well at school or college.	T1	33.8	31.0
	T2	30.2	17.9

ME-WE FTF = face-to-face ME-WE intervention; ME-WE APP = app-based ME-WE intervention.

**Table 9 healthcare-12-02124-t009:** Summary content analysis of AYCs’ experiences of caring activities.

Question	Category	Summary of Findings	Illustrative Quote
Most positive caring activities and why they were perceived as positive	Emotional care	Mentioned as one of the most positive activities in all countries. Includes different aspects such as talking to the care recipient, keeping them company, walking together, etc. Helps the care recipient feel better and/or be happier, which contributes to the AYC’s own well-being.	*To help her to understand her own feelings and how to deal with toxic friendships. Make her happy. When I get her to laugh and be happy, when I got her to eat breakfast, and when I see that she’s fine, then I feel that I’ve succeeded, and I like myself.* (AYC, T1, SE)
	Domestic/Household tasks	Mentioned as one of the most positive activities in all countries. Helps the care recipient feel better, which contributes to the AYC’s own well-being. Also perceived by AYCs to be less demanding.	*I like it very much when I help my dad with cooking and baking. Last time we did plum dumplings. It was a relaxing atmosphere, we laughed a lot and enjoyed ourselves. I learned many new things. I also like watering plants and working in the garden very much*. (AYC, T1, SI)
	Taking care of siblings who are care recipients	Also mentioned as one of the most positive activities in all countries as it makes the sibling feel good and grow personally, which in turn contributes to the AYC’s own well-being.	*Supporting my sister emotionally being her person to confide in. Feel accomplished not just as a carer but also as a big sister.* (AYC, T0, UK)
Most negative caring activities and why they were perceived as negative	Emotional care	Also mentioned in all countries as one of the most negative caring activities, because it was deemed by some AYCs to be stressful and demanding if there is a lack of agreement between the AYC and the care recipient. Exhausting situations that make the AYC feel bad are, for example, trying to convince relatives not to drink alcohol, and constantly trying to prevent conflicts or self-injury.	*To constantly keep family members in check so they don’t quarrel, due to mental health problems. For example, with a mum with Alzheimer and a dad with PTSD, which gives him a hot temper […] Mum doesn’t understand when she should keep a low profile. Friction occurs frequently.* (AYC, T0, SE)
	Personal care	Mentioned in all countries (except in the UK). Considered to be hard/too demanding, stressful, or making the AYC feel uncomfortable. Doing things without reward or positive feedback also adds to the feeling of negativity.	*Assisting my aunt when she is in the bathroom and is nervous, because she doesn’t know how to manage.* (AYC, T0, IT).
	Domestic/Household tasks	Domestic tasks were mentioned as the most negative activity in all countries, while taking care of siblings was not mentioned as the most negative activity in the Netherlands or in the UK. Reasons were not always presented, but some AYCs mentioned disliking the activity if the activity interferes with other activities or if it is non-rewarding.	*I dislike always having to do lots of things around the house because sometimes it can interfere with my personal life outside of caring such as schoolwork and social aspects.* (AYC, T1, UK)
	Taking care of siblings		*It’s also hard when I have to look after my brother when he’s angry, because then I do my best, but I don’t get anything good in return.* (AYC, T1, NL)
Why caring activities are rewarding	The outcome of caregiving	Aspects related to the outcome of caregiving were reported to be most rewarding in all countries, as they help the care recipient, making them happy or having a good influence on them.	*When a lot is done and done successfully so much so that you have relieved some of the pressure the person you are caring for has been feeling good which then improves their well-being.* (AYC, T1, UK)
	The relationship between AYC and care recipient	Aspects related to the relationship between the AYC and the care recipient were also reported as rewarding in all countries (except in the Netherlands). Sometimes they were similar to the aspects of the outcome such as ‘helping people’, but aspects such as ‘good friendship’, ‘getting thanks’, or ‘being there for the care recipient’ were also reported.	*Helping my twin sister, because in this way our bonding gets stronger and we are building our relationship.* (AYC, T0, SI)
	The process of caring	The process of caring was mentioned as rewarding in all countries, and this was most common in Sweden and Slovenia.	*I’ve been able to learn new skills throughout my caring, e.g., cooking, gardening, trying different sports with my brother. The most rewarding was to help my brother learn to speak.* (AYC, T0, UK)
Why caring activities are upsetting	The process of caring	The most common reasons for caring activities being upsetting were aspects related to the process of caring. They were mentioned by most respondents in Italy, Sweden, Slovenia, the Netherlands, and the UK and concern aspects such as worrying about the care recipient and their illness, listening to depressive talk, or stressing personal caring tasks as very demanding.	*I get the most negative feelings when helping with personal hygiene because of the fear that I will not handle the situation, that I will cause more pain to my mother.* (AYC, T2, SI)
	The relationship between AYC and care recipient	Another common theme in all countries. Aspects such as the recipient being dependent on the carer, that no one cares for the AYC, and having to ignore their own feelings and needs are described as upsetting. The minimizing of their own needs and continuously adapting to the needs of others were mentioned by AYCs in the Netherlands.	*Having to hold back my emotions, and not showing that I’m sad when she is around. When she is sad and gets mean/doesn’t think about other people’s feelings.* (AYC, T0, SE)
	The outcome of caregiving	Aspects related to outcomes were mentioned in all countries, such as being out of control, no matter how hard you try.	*When my brother is angry. He hits me, he bites himself and his clothes and he shouts. Trying to calm him down, often doesn’t help.* (AYC, T1, NL)

**Table 10 healthcare-12-02124-t010:** Summary content analysis of AYCs’ experiences and perceived changes during the COVID-19 pandemic.

Question	Category	Summary of Findings	Illustrative Quote
Life changes during COVID-19 pandemic	Positive changes	The most common positive change was the experience of having more time for them-selves, for favorite activities, or for self-reflection. In addition, more time with the family was appreciated, making some of the AYCs more relaxed. Additionally, especially in Slovenia, online schooling was reported to make life less stressful and reduced the fear of bringing COVID-19 home to loved ones. Comparing T1 and T2, similar themes were put forward, but in T2, new insights and better relations were also reported.	*I’m more relaxed and calmer, I’m no longer nervous and tired, during the lock-down, I had a lot of time and I used it to get to know myself and spend more time with my family.* (AYC, T1, SI)
	Negative changes	Among the negative aspects, social isolation and not being able to meet up with friends were the most common. This, as well as an increased level of caring responsibility, the feeling of being left alone with the care recipient, and an increased worry for the care recipient when not being allowed to see them, affected the well-being and mental health of some AYCs. Likewise, loneliness and depression were reported. Comparing T1 and T2, similar themes were emphasized, but in T2, poorer school results and raised worries were also reported.	*During the pandemic you feel so alone, and you only have the person you care for. It feels like it is never ending and there is no light at the end of the tunnel while all you do is care and there is no space for you to think or breathe. Isolated from anyone or anything that keeps you going. Slowly suffocating in silence.* (AYC, T1, UK)
Needed and received support services during COVID-19	Needed support	The data on needed support were sparse and reported by fewer respondents. The majority of the AYCs reported that their needs for support had not increased, neither for themselves nor for their families. Only a few AYCs directly stated that they or their families’ need for support increased during the pandemic. In Slovenia, mostly financial and psychological support were outlined, while AYCs from the other countries did not specify what kind of support.	*I was thinking to turn again for some help from the psychotherapist because of my problems and distress, for which I thought before that I don’t need, but it is becoming more obvious that I need professional help.* (AYC, T1, SI)
	Received support	The responses were few and data scarce. Some AYCs reported that neither themselves, nor their families, received professional support, while others stated they had very good support. Some also stated that they received support from friends, family, and/or the ME-WE group.	*We had very good support during this time. It was nice to know that despite this all happening, there are people who are there for you and really want to be.* (AYC, T2, NL)
Changes in health and well-being due to COVID-19 pandemic	Positive changes	AYCs in all countries but Sweden reported both positive and negative changes in their overall well-being and mental health. Positive aspects were having more time for taking care of oneself, e.g., running regularly or eating healthier food, more time with their family, more sleep, and less stress.	*Life in the lock-down was in all senses less stressful. Still, I do not feel that the pandemic would in any way significantly affect my physical health. If something, I take better care of myself in a sense that I run more regularly than before, furthermore I sleep more, as I am aware that this is to certain extent a prevention from the infection with the coronavirus.* (AYC, T1, SI)
	Negative changes	Negative aspects included the lack of social life, more conflicts in the family that sometimes affected their mental health, with examples such as depression, feelings of loneliness, anxiety, and self-injury. Some AYCs were less physically active and gained weight. Some AYCs from the Netherlands also reported physical changes such as headache and fatigue. Sweden was the only country that solely reported negative changes.	*My mental health is very poor at the moment, but I think this is mainly because there has always been stress at home in recent years and I never re-ally had time for myself. Now, for the first time in years, things have been quiet for a few months and for the first time I have time to really process everything and have time for myself, but I don’t know if the pandemic has had an influence on this.* (AYC, T2, NL)

**Table 11 healthcare-12-02124-t011:** Summary content analysis of AYCs’ responses to the Post-Intervention Self-Assessment (PISA).

Question	Category	Summary of Findings	Illustrative Quote
Intervention outcomes	Help and support from the intervention	AYCs in both blended (SE, NL) and face-to-face delivery countries (IT, SI, UK) stated that they received help and support from the ME-WE intervention. In both delivery approaches, AYCs received help especially with dealing with stressful thoughts and feelings, learning more about themselves, and being kind to themselves.	*A bit of processing, and how you can put things off your chest. I don’t always man-age to do this very well yet, but these are certainly things that I will use more often.* (AYC, T1, NL)
	Changes in life as a result of the intervention	AYCs in all countries reported positive changes in their lives, such as handling stressful thoughts and feelings in a better way (mentioned in most countries), and the ability to be more forgiving and kinder to oneself and/or to take better care of oneself.	*I got to know myself better, all the facets of my character. I also learned the im-portance of giving myself time and started doing it more often.* (AYC, T1, IT)
	Changes in caring activities as a result of the intervention	AYCs in blended delivery countries (SE, NL) expressed a greater variation concerning changes in their caring activities due to the intervention compared to AYCs in face-to-face delivery countries (IT, SI, UK). The experience of receiving more support in the caring role stood out, which was only reported by AYCs in blended delivery countries (SE, NL) and not by AYCs in face-to-face delivery countries (IT, SI, UK). Decreased levels of caring and changed forms of caring activities were reported most frequently at T1. At T2, the most common change across all countries was the feeling of being more confident or capable as a carer.	*I’m starting to like it more and I know more about how to handle the situation.* (AYC, T2, SE) *Unpleasant caring activities I do less often. In this regard, I’m also more assertive: if I don’t want to do something or I feel uncomfortable with something, I say. Even if there’s no one else that could do this unpleasant task instead of me, it helps me, since it seems to me, that my mother as care recipient, as well as others, understand me better and are therefore more considerate in relation to me.* (AYC, T2, SI)
	Negative aspects of attending the intervention	In all countries, some negative aspects were raised, although most of the AYCs had no negative remarks about the intervention. The topics differed between the countries. One more common theme was the group sessions, which in some cases were experienced as intense, hard to cope with, and uncomfortable.	*At times it was hard to open up about things from my private life or actually see the facts about me that I tend to brush away or ignore.* (AYC, T1, UK)
Experiences of attending the intervention during COVID-19 pandemic	Positive aspects	Positive aspects of attending the meetings in general were most commonly mentioned, which related to other subcategories such as having contact with other AYCs, being positive to online meetings and helping to create meaning and routines during the pandemic.	*Workshops really had a positive influence on me, every week this was one of the most relaxing things. At the same time, I got a good feeling about myself and a feeling that I’m also able to help others, when they feel down. Considering that the workshops were also online, this didn’t bother me. I recognized that in these kinds of periods, we shouldn’t suppress our feelings, but release them and consequently become aware of them and after somehow slowly control them.* (AYC, T1, SI)
	Negative aspects	Negative aspects mostly concerned the online form, causing fatigue after a day of schooling online. Some AYCs also mentioned the home exercises, adding to the workload of school homework.	*I found it very stressful to also have this, especially when you have so little time.* (AYC, T2, NL)

**Table 12 healthcare-12-02124-t012:** Summary content analysis of the evaluation of the ME-WE app.

Question	Category	Summary of Findings	Illustrative Quote
Evaluation of the ME-WE app	Positive aspects	In Sweden, most of the AYCs stated that the app was helpful and supportive and that they would recommend it. In the Netherlands, AYCs commented that they used the app in preparation for the sessions, and one participant thought the app had a lot of potential once it was finished, and that it was intuitive to use.	*The app was not yet ready according to the plan at that time [of the intervention] and therefore not good to use. The app does have a lot of potential!* (AYC, T2, NL)
	Negative aspects	AYCs were most dissatisfied with the usability of the app. Most problems were technical and network issues. Most of the AYCs in the Netherlands stated that the app had not been helpful or supportive, and only half of them would recommend it.	*Because the app is in its infancy, not all parts are well developed yet. I have only used the app during sessions and did not find it useful yet. I think if there were more updates, I would get more out of it.* (AYC, T1, NL)

**Table 13 healthcare-12-02124-t013:** Summary content analysis of AYC participants’ feedback on the evaluation questionnaire.

Question	Category	Summary of Findings	Illustrative Quote
Views of the evaluation questionnaire	Positive aspects	The questions were described as a help for further reflection on their situation, and providing a feeling of being understood, or being helped.	*It took quite a bit of deep thoughts and almost emotional after seeing how I would have answered the questions before. I feel like I have changed as a person and the questionnaire just made a perfect conclusion to my experience.* (AYC, T2, UK)
	Negative aspects	Questions were perceived as confronting or triggering negative feelings or experiences. The survey was deemed by some AYCs to be lengthy and some questions difficult to understand.	*I’m feeling a little down in the dumps now because this questionnaire made me rethink my mom’s ill-ness and gives me a sense of fear for the future.* (AYC, T0, IT)

## Data Availability

The ME-WE project reports, deliverables and outputs are publicly accessible from the project website: https://me-we.eu (accessed on 15 July 2023). Data were submitted to the ADP data archive and have the following citation. Hlebec, V., Bolko, I., Magnusson, L., Brolin, R., Becker, S., Lewis, F., Morgan, V., Leu, A., Alder, E., Phelps, D., De Boer, A., Hoefman, R., Bouwman, T., de Jong, N., Santini, S., Socci, M., D’Amen, B. and Boccaletti, L., Hanson, E., (2024). Psychosocial support for promoting mental health and well-being among adolescent young carers in Europe (ME-WE), 2021 [Data file]. Ljubljana: Univerza v Ljubljani = University of Ljubljana, Arhiv družboslovnih podatkov = Slovenian Social Science Data Archives. ADP—IDNo: MEWE21. https://doi.org/10.17898/ADP_MEWE21_V1.

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
