# Peer review of "Promoting Mental Health and Well-Being Among Adolescent Young Carers in Europe: A Cross-National Randomized Controlled Trial Study"

_healthcare, 2024, doi:10.3390/healthcare12212124_

Round 1

Reviewer 1 Report

Comments and Suggestions for Authors

The article is very interesting and well-written. The statistics are clear and comprehensible, and the recommendations are articulated clearly and thoughtfully. The qualitative section is also very clearly written. I recommend publishing the article after minor revisions.

To enhance the paper, you could consider adding two significant articles that were written during the COVID-19 pandemic:

  • https://pubmed.ncbi.nlm.nih.gov/37658247/
  • https://www.sciencedirect.com/science/article/abs/pii/S088259631930630X
  • https://www.ncbi.nlm.nih.gov/pmc/articles/PMC8412871/

Additional questions to consider:

  • Did you conduct factor analysis on the questionnaires you distributed? What were the results of the factor analysis?
  • Did you obtain ethical approval for each country where the research was conducted?
  • Did the children aged 15-17 sign consent forms to participate in the research, or did their parents sign the ethical approval?
  • Was there a difference in the effectiveness of Zoom-based training compared to face-to-face training?
  • Is it recommended to have the article linguistically edited for English?

These questions could add depth and important details to the research and its findings.

Comments on the Quality of English Language

The article is very interesting and well-written. The statistics are clear and comprehensible, and the recommendations are articulated clearly and thoughtfully. The qualitative section is also very clearly written. I recommend publishing the article after minor revisions.

To enhance the paper, you could consider adding two significant articles that were written during the COVID-19 pandemic:

  • https://pubmed.ncbi.nlm.nih.gov/37658247/
  • https://www.sciencedirect.com/science/article/abs/pii/S088259631930630X
  • https://www.ncbi.nlm.nih.gov/pmc/articles/PMC8412871/

Additional questions to consider:

  • Did you conduct factor analysis on the questionnaires you distributed? What were the results of the factor analysis?
  • Did you obtain ethical approval for each country where the research was conducted?
  • Did the children aged 15-17 sign consent forms to participate in the research, or did their parents sign the ethical approval?
  • Was there a difference in the effectiveness of Zoom-based training compared to face-to-face training?
  • Is it recommended to have the article linguistically edited for English?

These questions could add depth and important details to the research and its findings.

Author Response

Reviewer 1:

The article is very interesting and well-written. The statistics are clear and comprehensible, and the recommendations are articulated clearly and thoughtfully. The qualitative section is also very clearly written. I recommend publishing the article after minor revisions.

To enhance the paper, you could consider adding two significant articles that were written during the COVID-19 pandemic:

  • https://pubmed.ncbi.nlm.nih.gov/37658247/
  • https://www.sciencedirect.com/science/article/abs/pii/S088259631930630X
  • https://www.ncbi.nlm.nih.gov/pmc/articles/PMC8412871/

Answer: Thank you very much for your thoughtful review and for suggesting additional references to enhance the manuscript. We have carefully considered the articles you recommended. While we greatly appreciate the value of these works, we decided not to include them in the manuscript as they do not specifically relate to the field of caregiving at young ages. Including these references would extend the scope beyond the focus of our paper, which is adolescent young carers. Our aim is to maintain a concentrated reference list aligned with the core themes we are exploring, and we believe the current literature cited is sufficient to support the points made in our manuscript.

Additional questions to consider:

Did you conduct factor analysis on the questionnaires you distributed? What were the results of the factor analysis?

Answer: Thank you for your valuable feedback and for inquiring about the factor analysis of the questionnaires. Unfortunately, the sample sizes for each country-specific subsample were too small for a robust testing of structural validity through factor analysis. However, the measures used in our study were selected based on their validation history with adolescent samples, and we ensured rigorous translation and adaptation procedures in the participating countries where a validated version was not already available. Importantly, we computed and reported reliability coefficients for each measure within each country, and the results were consistently within acceptable ranges, confirming the internal consistency of the scales used. We believe this supports the reliable application of scale scores across our diverse sample. We have acknowledged the lack of structural validity testing and clarified our approach in the Limitations section (see page 29, lines 939-949).

  • Did you obtain ethical approval for each country where the research was conducted

Answer: Yes, we can confirm that formal ethical approval was obtained in five of the six countries (IT, SI, NL, UK and SE) in keeping with national human ethics legislation. Formal ethics approval was not deemed necessary in CH by the relevant Swiss ethics review body, in keeping with their national human research ethics legislation. However, detailed opinions were sought and secured from the Swiss ethics review body by the Swiss team. The text in the main body of the paper regarding ethical approval has now been revised to make it clearer for the reader (see page 4 from line 177 onwards) and the reader is also directed to the Institutional Review Board Statement at the end of the paper.

  • Did the children aged 15-17 sign consent forms to participate in the research, or did their parents sign the ethical approval?

Answer: Yes, we can confirm that all children/young people who were enrolled to the study, signed consent forms to participate in the research. This is now clearly included in the main body of the paper together with information about parental consent. See the newly included paragraph immediately following the ethical approval paragraph (pages 4-5, line 202 onwards). Please note that parental consent procedures varied according to the legal age parental consent is required for young people and/or according to national good ethical practice existing in the six partner countries (see p4, from line 206 onwards).

  • Was there a difference in the effectiveness of Zoom-based training compared to face-to-face training?

Answer: The FTF and APP intervention groups are already tested in the mixed ANOVA. The interaction effect examines whether changes in mean scores across time were the same across all groups. A nonsignificant time*interaction indicates that the two intervention groups and the control group showed equal changes. Any further statistical examination is not possible owing to intertwined initial two delivery modes (FTF and APP) with zoom delivery which was endorsed owing to social distancing measures and to the small sample sizes.

  • Is it recommended to have the article linguistically edited for English?

Answer: As three of the co-authors, including the final author, are native English speakers and the final author has now duly carried out an additional final language edit of the entire manuscript, we consider that it is not necessary to have the current revised version of the article linguistically edited for English.

  • These questions could add depth and important details to the research and its findings.

Reviewer 2 Report

Comments and Suggestions for Authors

The topic of the manuscript "Promoting Mental Health and Well-Being among Adolescent Young Carers in Europe: A Cross-National Randomized Controlled Trial Study" is interesting. However, the manuscript is very confusing. The following are some of the deficiencies found.

-In the abstract, page 1, lines 44-45, it says "We designed a randomized controlled trial with 217 AYCs participating in the study, either in the intervention or control group". But this number is higher than the number of participants in the study shown in Figure 1 (page 4).

-The Introduction section should be expanded as it does not provide a complete rationale for the work performed. For example, it should state why they opted for the intervention implemented, and its advantages and/or disadvantages compared to other alternatives.

-In the introduction on page 2, lines 68 and 69 it says "the primary prevention interventions for AYCs" and on the same page, lines 87 and 88, it says "primary intervention". It should be clear whether if it is a prevention program or not.

-The Materials and Methods section, page 3, lines 111 to 116, states: "Two delivery methods were applied for two groups of countries in the RCT. The main difference between the delivery modes was the use of only face-to-face methods or also online methods for facilitating sessions were participants met. The first mode consisted of a fully face-to-face approach (FTF) and was implemented in the three countries (Italy, Slovenia, and United Kingdom). The second delivery mode combined face-to-face methods with an online tool for facilitating training sessions”. The differences between the two delivery methods should be clearly described, as well as what the online tool consisted of. Although in subsection “2.3. Intervention” explains more details of the intervention program, the differences are still unclear. In addition, the reasons why two methods were used should be explained.

-On Page 5, line 201 it says "4 AYCs (1 AYC in the intervention and 3 AYCs in the waitlist group) in Switzerland". The fact that there is only one participant from Switzerland in the intervention program calls into question whether that country should be included as a participant in the study.

-On page 5, lines 203 to 205, it says "“Intervention group membership was reduced by excluding participants who attended less than 50 % of the sessions prior to the statistical analyses (and considered dropouts and hence not included in the above reported sample sizes)." The number of participants who attended less than 50% of the sessions should be specified.

-On page 5, lines 206 to 208 it says: "Differences across countries were sizable, however a common tread was that recruitment for control groups was often delayed, and they contain smaller number of participants than intervention groups." This sentence should be revised because this is not the case in all countries or even globally.

-The subsection "2.4. Study Outcomes" should be revised and the measures should be made clearer.

-On page 7 on lines 309 to 313 it states "Caring-related quality of life was evaluated using three ad-hoc closed-ended questions (Yes/No) about thoughts of self-harm and harming others, being bullied, teased or made fun of, and a multiple-choice question addressing health-related issues resulting from the caring role (i.e., mental health problems, physical health problems, or other health-related conditions)." This description is very confusing, especially the multiple-choice question.

-On page 7 on lines 298 to 301 it states "Subjective mental health was assessed using the Warwick Edinburgh Mental Well-Being Scale (WEMWBS) [27]. The WEMWBS includes 14 items rated on a 5-point 299 scale from ‘none of the time’ to ‘all of the time”. This questionnaire seems unsuitable for assessing mental health as it focuses on the study of mental well-being.

- In the Results section on page 11 is "Table 2. Characteristics of Participants' Family Members Who Needed Help or Support". Except for the Me-We FTF, the number of people listed in each treatment condition "Me-We APP (n = 47)" "Waiting list (n = 125)" appears to be higher than the number of people listed elsewhere in the text, including Table 1. "Sociodemographic characteristics of both intervention groups and waitlist control group with corresponding test statistics" where the number of people in the Me-We APP is 32 and in the waitlist is 106.

- Table 4 and the comments in the text (pages 12 to 14) should be revised and better explained, as they are difficult to understand.

Author Response

Review 2:

The topic of the manuscript "Promoting Mental Health and Well-Being among Adolescent Young Carers in Europe: A Cross-National Randomized Controlled Trial Study" is interesting. However, the manuscript is very confusing. The following are some of the deficiencies found.

-In the abstract, page 1, lines 44-45, it says "We designed a randomized controlled trial with 217 AYCs participating in the study, either in the intervention or control group". But this number is higher than the number of participants in the study shown in Figure 1 (page 4

Answer:  We can clarify that the number of participants in Figure 1 adds up to 213. The missing 4 are Swiss participants who are not included in the figure (see Footnote e). We have now duly added an additional explanation to the text. See page 6, line 226, page 6 lines 266-269.

-The Introduction section should be expanded as it does not provide a complete rationale for the work performed. For example, it should state why they opted for the intervention implemented, and its advantages and/or disadvantages compared to other alternatives.

Answer: Thank you for encouraging us to elaborate on presenting the intervention and fully describe the rationale for choosing this one against others.  We have now duly expanded the Introduction section accordingly to provide a more comprehensive rationale for the work performed (see page 2 lines 75-98).

-In the introduction on page 2, lines 68 and 69 it says "the primary prevention interventions for AYCs" and on the same page, lines 87 and 88, it says "primary intervention". It should be clear whether if it is a prevention program or not.

Answer: Thank you for spotting this error, lines 87 and 88 have now been corrected so that it also reads “primary prevention intervention” here too.

-The Materials and Methods section, page 3, lines 111 to 116, states: "Two delivery methods were applied for two groups of countries in the RCT. The main difference between the delivery modes was the use of only face-to-face methods or also online methods for facilitating sessions were participants met. The first mode consisted of a fully face-to-face approach (FTF) and was implemented in the three countries (Italy, Slovenia, and United Kingdom). The second delivery mode combined face-to-face methods with an online tool for facilitating training sessions”. The differences between the two delivery methods should be clearly described, as well as what the online tool consisted of. Although in subsection “2.3. Intervention” explains more details of the intervention program, the differences are still unclear. In addition, the reasons why two methods were used should be explained.

Answer: the two delivery modes for our intervention are now described in more detail and we hope that they are now easier to understand for the reader. See page 3, lines 132-148.

-On Page 5, line 201 it says "4 AYCs (1 AYC in the intervention and 3 AYCs in the waitlist group) in Switzerland". The fact that there is only one participant from Switzerland in the intervention program calls into question whether that country should be included as a participant in the study.

Answer: Thank you for this comment. Switzerland faced major recruitment challenges. The sole AYC participant in the intervention group actively participated in all the seven ME-WE sessions with other young carers who did not meet the inclusion criteria (compassionate cases who were slightly younger or older than the target group). Due to the small number of participants in Switzerland, the few data were not included in the international quantitative analysis (see additional text page 11 lines 495-497). However, the Swiss qualitative findings have been included in the analysis. First of all, for the qualitative part of the study we were interested in AYC’s experiences of caring activities, their experiences during the COVID-19 pandemic and the evaluation of the ME-WE app independently of the country in which they lived. Additionally, we considered these qualitative data to be highly informative in order to gain information on the recruitment challenges experienced in Switzerland, partly due to the changes during the COVID-19 pandemic, but also due to the experiences in their caring role.

-On page 5, lines 203 to 205, it says "“Intervention group membership was reduced by excluding participants who attended less than 50 % of the sessions prior to the statistical analyses (and considered dropouts and hence not included in the above reported sample sizes)." The number of participants who attended less than 50% of the sessions should be specified.

 -On page 5, lines 206 to 208 it says: "Differences across countries were sizable, however a common tread was that recruitment for control groups was often delayed, and they contain smaller number of participants than intervention groups." This sentence should be revised because this is not the case in all countries or even globally.

Answer: thank you for your comment, the sentence has now been re-worded accordingly, see  pages 6-7,  275-277:  

“Differences across countries were sizable, however a common tread was that recruitment for control groups was often delayed and resulted in uneven distribution of participants in the intervention and control groups”.

-The subsection "2.4. Study Outcomes" should be revised and the measures should be made clearer.

Answer: We have now duly added additional information to make the measures clearer. See pages 8-10.

-On page 7 on lines 309 to 313 it states "Caring-related quality of life was evaluated using three ad-hoc closed-ended questions (Yes/No) about thoughts of self-harm and harming others, being bullied, teased or made fun of, and a multiple-choice question addressing health-related issues resulting from the caring role (i.e., mental health problems, physical health problems, or other health-related conditions)." This description is very confusing, especially the multiple-choice question.

-On page 7 on lines 298 to 301 it states "Subjective mental health was assessed using the Warwick Edinburgh Mental Well-Being Scale (WEMWBS) [27]. The WEMWBS includes 14 items rated on a 5-point 299 scale from ‘none of the time’ to ‘all of the time”. This questionnaire seems unsuitable for assessing mental health as it focuses on the study of mental well-being.

- In the Results section on page 11 is "Table 2. Characteristics of Participants' Family Members Who Needed Help or Support". Except for the Me-We FTF, the number of people listed in each treatment condition "Me-We APP (n = 47)" "Waiting list (n = 125)" appears to be higher than the number of people listed elsewhere in the text, including Table 1. "Sociodemographic characteristics of both intervention groups and waitlist control group with corresponding test statistics" where the number of people in the Me-We APP is 32 and in the waitlist is 106.

Answer: Thank you for pointing this out. These are not characteristics of participants described in these two tables but characteristics of people that AYCs are caring for. 

Each participant/ AYC had to name the number of family members and/or close friends that they provide support to and answer some questions about these people. The number of these people named was summed up and their characteristics are presented in Tables 2 and 3.

Hence, the numbers in Tables 2 and 3 do not match Table 1 - in the first case we have AYCs in the other two we have people that AYCs are providing care and support to.

 (p12, line 535)

- Table 4 and the comments in the text (pages 12 to 14) should be revised and better explained, as they are difficult to understand.

Answer: Thank you for pointing this out. We have now added headings for the columns in Table 4 and revised the description of the results to enhance clarity. See pages 15-16.

Reviewer 3 Report

Comments and Suggestions for Authors

Dear authors and editor,
 Congratulations on your manuscript.

Despite the time interval between the study and the time of publication, it contains new ideas and findings and makes new contributions to the literature, taking into account the global conjecture.
 We humbly leave some suggestions in comments on the PDF that can help you reflect better on some issues and suggest clarifying in depth the implications for practice, research, education and global/international policies.

Many congratulations on your manuscript and huge success for publication and future studies.
 We share with you just a few comments (attached).
 Sincerely

Comments on the Quality of English Language

Very understandable, minor editing of English language required.

Author Response

R3:

Thank you very much! All your edits were accepted. Also, final author has now duly carried out an additional final language edit of the entire manuscript to prevent any mistakes.

Keywords – term suggested by template

Removed comma L63, removed space, L65

Publication was delayed for several unrelated, but unfortunate, circumstances. First, the field work faced severe delays owing to challenging recruitment, then by COVID-19 preventive measures. This lead to extended, but not financed period of project duration. National funds were secured for 6 months after the project was completed but was not enough to complete upgraded analysis for the purposes of this publication. Further delay was faced when extra national funding has fell through and we were therefore faced with completion of the manuscript on voluntary, not funded, basis.

Group added L126

Comma deleted L274

Pannoc description elaborated L413-421

PISA – L433 corrected and systematically changed throughout the manuscript

Two “the” added L975

Last, but not least deleted L1032

Round 2

Reviewer 2 Report

Comments and Suggestions for Authors

The revised version of the manuscript “Promoting Mental Health and Well-Being among Adolescent Young Carers in Europe: A Cross-National Randomized Controlled Trial Study” has been improved substantially compared to the original version.

However, the manuscript should be thoroughly revised as some deficiencies have been identified that should be corrected prior to publication. For example, the KIDSCREEN-10 and HBSC questionnaires are missing from the footer of Table 4 (pages 15 to 16, lines 599 to 603).

- Page 2, line 79 states “that one of the most promising models for use with YCs was The Resourceful Adolescent”

YCs should be defined as its definition does not appear, only AYCs and it is not clear if YCs also refers to adolescents. (YCs also appears on page 3, line 134).

-Several typos have been noted. For example, on page 2, line 74, it says "ME-WE primary prevention intervention programis aimed at promoting mental health". There should be a space between program and is. And on page 3, line 129, it says "The first mode, nameda fully face-to-face approach (FTF), was". The word "nameda" should be corrected.

Author Response

R2:

However, the manuscript should be thoroughly revised as some deficiencies have been identified that should be corrected prior to publication. For example, the KIDSCREEN-10 and HBSC questionnaires are missing from the footer of Table 4 (pages 15 to 16, lines 599 to 603).

Answer: Thank you very much for this comment. The footer is now duly corrected. (L656-658)

- Page 2, line 79 states “that one of the most promising models for use with YCs was The Resourceful Adolescent”

Answer: Thank you for spotting this. The YCs is replaced with AYCs.

YCs should be defined as its definition does not appear, only AYCs and it is not clear if YCs also refers to adolescents. (YCs also appears on page 3, line 134).

Answer: Thank you for spotting the YCs, which is now amended on page 3 line 145. The only difference between the YCs and AYCs is the age bracket, so the AYCs are a subgroup of YCs.

Several typos have been noted. For example, on page 2, line 74, it says "ME-WE primary prevention intervention programis aimed at promoting mental health". There should be a space between program and is. And on page 3, line 129, it says "The first mode, nameda fully face-to-face approach (FTF), was". The word "nameda" should be corrected.

Answer: Our apologies and thank you very much for spotting these typos -  they are all now duly corrected.